# Voltage-gated Na$^+$ currents in human dorsal root ganglion neurons

Xiulin Zhang[1], Birgit T Priest[2], Inna Belfer[3], Michael S Gold[4]*

[1]Department of Urology, The Second Hospital of Shandong University, Jinan Shi, China; [2]Lilly Research Laboratories, Indianapolis, United States; [3]Office of Research on Women's Health, National Institutes of Health, Bethesda, United States; [4]Department of Neurobiology, University of Pittsburgh School of Medicine, Pittsburgh, United States

**Abstract** Available evidence indicates voltage-gated Na$^+$ channels (VGSCs) in peripheral sensory neurons are essential for the pain and hypersensitivity associated with tissue injury. However, our understanding of the biophysical and pharmacological properties of the channels in sensory neurons is largely based on the study of heterologous systems or rodent tissue, despite evidence that both expression systems and species differences influence these properties. Therefore, we sought to determine the extent to which the biophysical and pharmacological properties of VGSCs were comparable in rat and human sensory neurons. Whole cell patch clamp techniques were used to study Na$^+$ currents in acutely dissociated neurons from human and rat. Our results indicate that while the two major current types, generally referred to as tetrodotoxin (TTX)-sensitive and TTX-resistant were qualitatively similar in neurons from rats and humans, there were several differences that have important implications for drug development as well as our understanding of pain mechanisms.

*For correspondence: msg22@ pitt.edu

## Introduction

It has long been appreciated that voltage-gated Na$^+$ channels (VGSCs) underlie the upstroke of the action potential, and therefore play an essential role in the propagation of action potentials along axons (*Hodgkin and Huxley, 1952a*, *1952b*). VGSCs consist of an alpha subunit, responsible for all essential features of a functional channel and up to two beta subunits that influence channel density and/or gating properties (*Yu et al., 2005*). Nine alpha subunits and four beta subunits have been identified. VGSCs have remained a target for the development of novel therapeutics because it is now also appreciated that changes in the biophysical properties (*Cantrell and Catterall, 2001*), distribution (*Cusdin et al., 2008*; *Kuba et al., 2010*), and/or expression (*Aptowicz et al., 2004*; *Qiao et al., 2013*) of these channels contributes to the dynamic regulation of neuronal excitability. Such changes have been particularly well documented in the context of pain, where both phosphorylation-dependent increases in Na$^+$ current, as well as more persistent increases in VGSC expression, have been shown to underlie both the acute and persistent increases in nociceptor excitability associated with inflammation (*Gold et al., 1996*; *Gould et al., 1998*). Similarly, both the ongoing pain and hypersensitivity associated with peripheral nerve injury are associated with changes in the pattern of VGSC expression (*Gold et al., 2003*; *Waxman et al., 1994*; *Hunter et al., 1997*) as well as distribution of channels in peripheral nerves (*Gold et al., 2003*; *Henry et al., 2007*; *Tseng et al., 2014*). From a therapeutic perspective, what has been particularly exciting about the evidence implicating VGSCs in inflammatory and neuropathic pain, is that several of the VGSC alpha subunits shown to contribute to the injury-induced increases in afferent excitability are preferentially expressed in the peripheral nervous system in general, and nociceptive afferents in particular

(*Gold and Gebhart, 2010*). This has raised the intriguing possibility that a VGSC subtype specific blocker would provide effective pain relief with minimal side effects.

Evidence in support of a role of VGSCs in a variety of pain states obtained with pre-clinical, largely rodent models, has been confirmed in pain patients. Similarly, genetic and anatomical evidence suggests that the VGSC alpha and beta subunits, and their pattern of expression, are similar in rodents and man. Nevertheless, virtually all that is known about the biophysical and pharmacological properties of VGSCs comes from the study of these channels in heterologous expression systems and in isolated rodent sensory neurons. The potential problem with this situation is highlighted by evidence that both the biophysical and pharmacological properties of channels are influenced by the expression system and species differences. For example, co-expression of VGSC beta1 and beta2-subunits with the alpha subunit NaV1.2 in frog oocytes not only increases current density, but the rate of current inactivation (*Patton et al., 1994*). In contrast, these beta-subunits have no influence on either current density or inactivation when co-expressed with NaV1.2 in tsA-201 cells (*Qu et al., 2001*). Rather, in tsA-201 cells, the beta-subunits drive a rightward shift in the voltage-dependence of channel activation. Similarly, the putatively NaV1.8 selective blocker A-803467 is over three orders of magnitude more potent against heterologously expressed human NaV1.8 than four of the other nine human alpha subunits tested, yet is only half as potent against the current believed to reflect activation of NaV1.8 channels, natively expressed in rat dorsal root ganglion neurons (*Jarvis et al., 2007*). Thus, the purpose of the present study was to determine the extent to which the biophysical and pharmacological properties of VGSCs described in rodent sensory neurons reflect the properties of the VGSCs in human sensory neurons. We focused on the two major classes of current that have been most extensively studied in rodent sensory neurons, those historically referred to as the low threshold rapidly activating, rapidly inactivating tetrodotoxin (TTX) sensitive current, and the high threshold more slowly activating and slowly inactivating TTX-resistant current, although voltage-protocols were used to isolate the putative TTX-resistant current in the majority of experiments described.

Whole cell patch clamp techniques were used to study acutely dissociated dorsal root ganglion (DRG) neurons from rats and humans. Our results suggest that while the two major $Na^+$ current types in neurons from rats are similar to those in neurons from humans, there are several potentially important pharmacological and biophysical differences that could contribute to the limited success in the development of novel pain therapeutics.

## Results

Neurons included in this study were obtained from 21 donors (13 males and 8 females). The average age of the donors was 45.2 years with a range of 13 to 77. Additional demographic data are summarized in *Table 1*. At least one data point was obtained from a total of 226 neurons that met inclusion criteria for clamp control and holding current. In order to maximize the amount of data collected from each neuron, we first determined whether it was possible to use voltage-clamp protocols to isolate the rapidly activating rapidly inactivating putative TTX sensitive (TTX-S) $Na^+$ current from the more slowly activating and slowly inactivating TTX-resistant (TTX-R) $Na^+$ current, as we have previously done in the rat (*Gold et al., 2003*). We subsequently confirmed that currents isolated in this manner were identical to those isolated with TTX (*Figure 1*). Based on these results, we refer to the slowly activating and slowly inactivation current resistant to steady-state inactivation as TTX-R current even though voltage steps rather than TTX was used to isolate this current in all subsequent experiments.

**Table 1.** Donor demographics and Na+ current density.

| Sex | Age (yrs) | TTX-R INa Current Density (pA/pF) | TTX-S INa Current Density (pA/pF) |
| --- | --- | --- | --- |
| Male (n = 13) | 40.9 ± 4.9 (1 Latin American, 1 African American, 11 Caucasian ) | −42.5 ± 2.7 (n = 69) | −66.2 ± 5.0 (n = 51) |
| Female (n = 8) | 52.3 ± 5.4 (all Caucasian) | −46.1 ± 5.3 (n = 50) | −54.4 ± 8.2 (n = 31) |

Data are mean ± SEM. Differences between males and females are not statistically significant ($p > 0.05$).

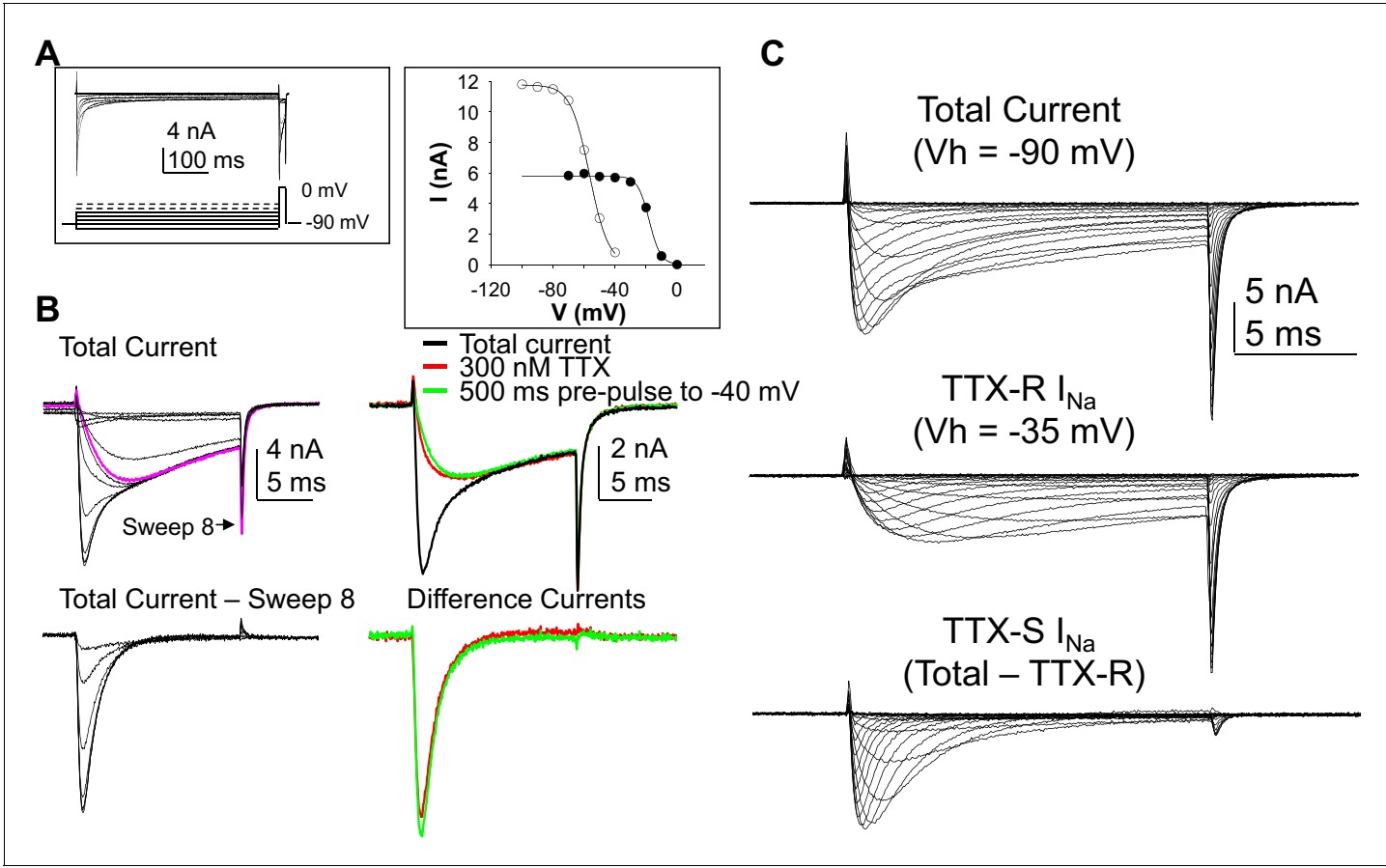

**Figure 1.** Separation of tetrodotoxin (TTX) sensitive (TTX-S) and resistant (TTX-R) voltage gated Na$^+$ currents in human DRG neurons. (A) A steady-state availability protocol was used to assess the voltage-dependence of inactivation of TTX-S and TTX-R currents. Left panel: The protocol consisted of a 500 ms pre-pulse to potentials between −100 and +10 mV, followed by test pulse to 0 mV. Right panel: The fast component of the current evoked at 0 mV (open circles), was inactivated over a range of test potentials more hyperpolarized than the range of test potential over which the slow component of the current evoked at 0 mV (closed circles) was inactivated. (B) Left Panel, top traces: Current evoked during the test pulse in A (left panel), plotted on a shorter time scale to more clearly illustrate the fast and slow components of the current evoked at 0 mV. Because the fast component of the current was completely inactivated with a test pulse more hyperpolarized than that at which the slow component began to inactivate, it was possible to digitally isolate the fast component by subtracting the slow component (purple trace), from the total current. Conversely, because the fast component was completely inactivated within ~10 ms of the start of the test pulse, it was possible to generate an availability curve for the slow component across the entire range of pre-pulse potentials. The bottom traces are those of the fast component digitally isolated from the slow component. Right panel, top traces: In another neuron, the steady-state availability protocol used in A, was used to inactivate the fast component of the current evoked at 0 mV. Application of 300 nM TTX removed the same component of the total current as the test pulse to −40 mV. Bottom traces: The difference between the total current and the current evoked at −40 mV, or that evoked in the presence of 300 mM TTX is virtually identical. (C). Current-voltage (I-V) protocols were used to assess current activation, with pre-pulse potentials that were based on steady-state availability data. Thus, total current (top traces) was evoked following a 500 ms pre-pulse to a potential at which currents were fully available for activation (i.e., −100 mV). TTX-R currents (middle traces) were evoked following a 500 ms pre-pulse to a potential at which TTX-S currents were completely inactivated, but TTX-R currents were fully available for activation (i.e, −35 mV). It was then possible to digitally isolate TTX-S currents (Bottom traces) from the total current by subtracting TTX-R current from the total current.

The following source data is available for figure 1:

**Source data 1.** Data plotted in *Figure 1*.

We initially focused on the small to medium diameter DRG neurons from both human and rat in this study for two main reasons. First, because the study of Na$^+$ channels in sensory neurons has largely been in the context of pain. In this context, data from guinea pig and rodents suggest that neurons with a small to medium cell body diameter are more likely to give rise to slowly conducting

axons (*Lawson, 2002*), which are, in turn, more likely to be nociceptive. Second, it was more difficult to maintain clamp control over the currents evoked from human sensory neurons with a larger cell body diameter. Our decision about the size range in which to consider a human DRG neuron small- or medium-diameter was based on the distribution of cell body sizes observed in crysections of whole ganglia obtained from three donors (*Figure 2A*, inset). Interestingly, in contrast to the skewed, clearly bimodal distribution of DRG neuron cell body sizes previously described in rodents (i.e., see [*Lawson et al., 1993*]), human DRG neuron cell body sizes were relatively normally distributed. We therefore included a subpopulation of larger neurons in subsequent experiments to further test the association between cell body diameter and phenotype in human DRG neurons. The average membrane capacitance of all the neurons included in this study was 116 pF (or ~60 ± 0.8 µm in diameter), with a range from 23 to 255 pF (or 26 to 90 µm). A histogram of the size distribution is plotted in *Figure 2*. As expected, human DRG neurons are significantly larger than small to medium (cell body diameter < 35 µm) rat DRG neurons, which had an average cell body capacitance of 36 pF, with a range of 22 to 70 pF (*Figure 2A*). Consistent with previous evidence from the rat suggesting that TTX-R currents are enriched in nociceptive afferents (*Djouhri et al., 2003*), which tend to have a small cell body diameter (*Lawson, 2002*), no TTX-R current was detected in rat neurons with a cell body capacitance >55 pF. Conversely, there was no correlation (p>0.05) between human DRG neuron cell body diameter and TTX-R current density, or the ratio of TTX-S to TTX-R current, and TTX-R currents were present in the largest neuron studied. Furthermore, the average capacitance of the nine neurons studied in which only TTX-S current was detected, 83.2 ± 18 pF, was, if anything, smaller (p=0.059), than that of the population of neurons with both TTX-S and TTX-R currents.

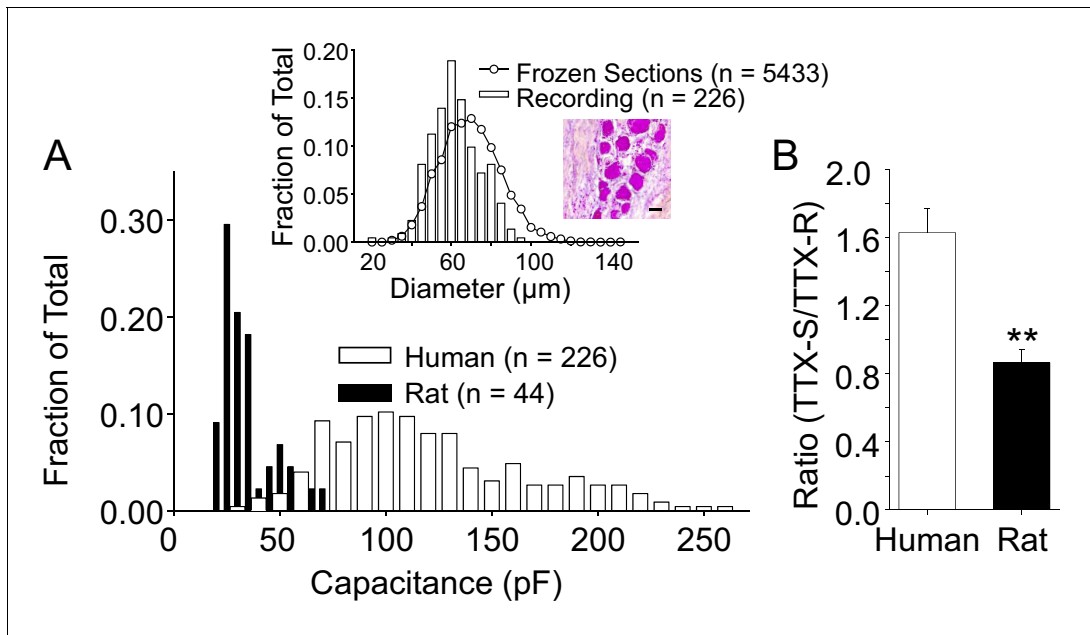

**Figure 2.** The size distribution of human and rat DRG neurons. (**A**) A histogram of the size distribution of all of the human (n = 226) and rat (n = 44) DRG neurons included in this study. Membrane capacitance was used as an indirect measure of cell size, since the capacitance is a reflection of the cell surface area. Capacitance was determined with amplifier circuitry. The average membrane capacitance of human DRG neurons was 116 pF, while that of rat DRG neurons was 36 pF. Inset: The cell size distribution of human DRG neurons from cryosections (20 µm) of paraformaldehyde post-fixed DRG. Data are from 20 sections, collected at 200 µm intervals, from each of three donors. Data from neurons studied with patch-clamp (Recording) have been replotted for comparison, where membrane capacitance was used to estimate cell body diameter based on the assumption that the capacitance of human DRG neurons is one µF/cm$^2$. An example of the tissue counted for this analyses is shown, where the scale bar is 50 µm. (**B**) In the neurons studied in which peak TTX-S and TTX-R were determined, the ratio of TTX-S to TTX-R current density was significantly greater in human (n = 71) than rat (n = 26) DRG neurons. ** is p<0.01.

The following source data is available for figure 2:

**Source data 1.** Data plotted in *Figure 2*.

Despite normalizing for the differences in cell body capacitance, the density of both TTX-R and TTX-S current were significantly larger in neurons from human ($-53.2 \pm 6.8$ pA/pF (n = 114) and $-62.2 \pm 4.3$ pA/pF (n = 78), respectively) compared to those in rat neurons ($-47.7 \pm 3.3$ pA/pF (n = 26) and $-42.6 \pm 4.6$ pA/pF (n = 26), respectively). Furthermore, the ratio of TTX-S to TTX-R current density was significantly greater in human than rat DRG neurons (*Figure 2B*).

## Inflammatory mediator-induced increase in both TTX-R and TTX-S currents in human DRG neurons

As the acute inflammatory mediator-induced potentiation of TTX-R currents in rat DRG neurons was one of the first observations driving a focus on these currents in the context of pain (*Gold et al., 1996*), we sought to determine the impact of inflammatory mediators on Na$^+$ currents in human DRG neurons. An inflammatory soup, consisting of bradykinin (10 μM), histamine (1 μM) and prostaglandin E2 (1 μM), was applied to neurons following establishment of stable recordings. Consistent with previous results from rat and subsequently mouse sensory neurons (*Yang and Gereau, 2004*), TTX-R currents were increased within seconds following bath application of inflammatory soup (*Figure 3*). This increase saturated within ~90 s. Comparable results were obtained in 13 of 17 human neurons tested, resulting in an average increase in peak current of $15 \pm 1\%$. Concomitantly, rates of current activation and inactivation were increased, $9 \pm 0.8\%$ and $12 \pm 3\%$, respectively. Both the

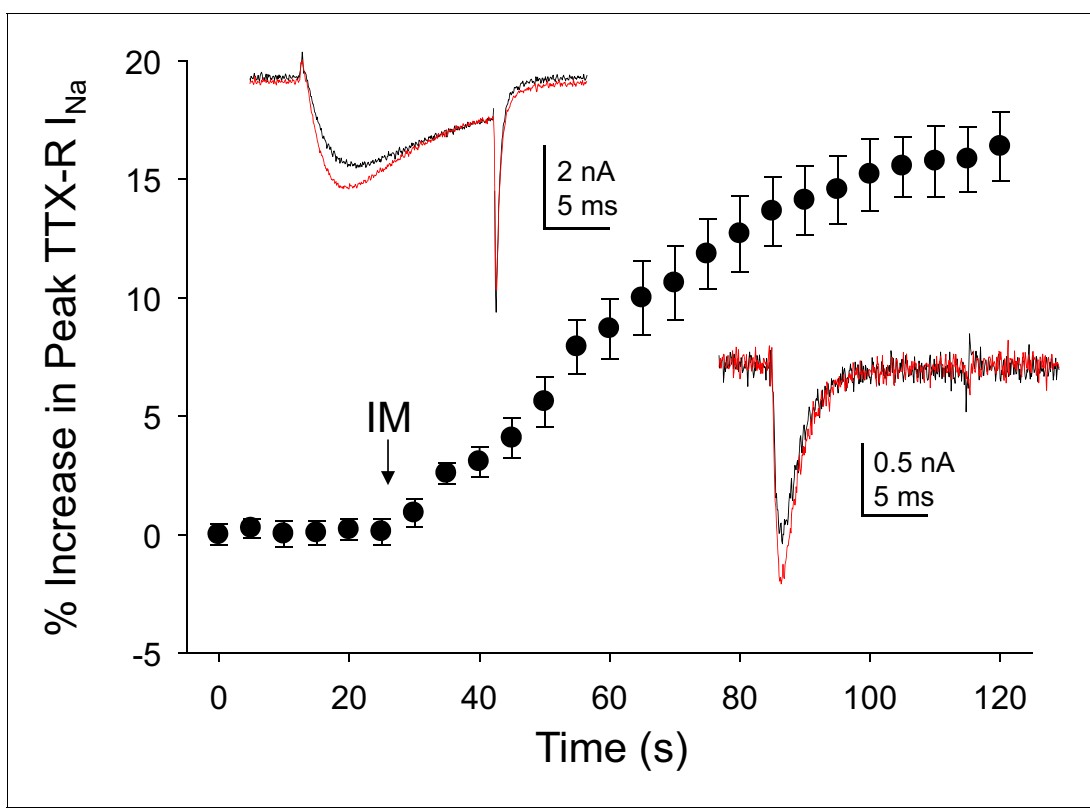

**Figure 3.** Inflammatory mediators (IM) increase Na$^+$ current in human DRG neurons. A combination of bradykinin (10 μM), histamine (1 μM) and prostaglandin E2 (1 μM) was bath applied to neurons after establishing the stability of evoked currents. (**A**) TTX-R currents were increased within seconds of IM application, and this increase was largely saturated within 90 s. In 13 of 17 neurons tested, the average increase in peak TTX-R current was $15 \pm 1\%$. TTX-S currents were increased by $30 \pm 6\%$ in four of 10 neurons tested. Insets: Typical traces of TTX-R (above) and TTX-S (below) current evoked with a voltage step to 0 mV before (black) and after (red) IM application, following a 500 ms prepulse to $-40$ mV or $-90$ mV, respectively.

The following source data is available for figure 3:

**Source data 1.** Data plotted in *Figure 3*.

increase in peak current and the increase in current activation rate would contribute to an increase in excitability.

Changes in TTX-S currents were assessed in 10 of the 17 neurons in which inflammatory soup-induced changes in TTX-R currents were assessed. TTX-S currents were increased in four of these neurons by $30 \pm 6\%$ (*Figure 3*). However, in contrast to TTX-R currents, the increase in TTX-S current was not associated with changes in rates of current activation or inactivation.

## Biophysical properties of TTX-R currents in rat and human DRG neurons

Because, as noted above, the vast majority of what is known about the biophysical and pharmacological properties of TTX-S and TTX-R currents in sensory neurons was derived from the study of rodent sensory neurons, we next sought to compare the biophysical and pharmacological properties of currents in human DRG neurons with those of currents in rat DRG neurons. Data were collected from DRG neurons obtained from eight rats under conditions identical to those used for the study of human DRG neurons with the same series of protocols. Data collection from these eight rats was interleaved with the collection of the last ~third of the human DRG neuron data, enabling the use of the same stock solutions of test reagents. The steady-state properties of TTX-R currents in rat DRG neurons were qualitatively similar to those in human DRG neurons. However, in contrast to previous results indicating that persistent TTX-R currents were larger in human DRG neurons (*Han et al., 2015*), we observed the opposite (*Figure 4A and B*), where the persistent current in rat neurons at −10 and 0 mV was significantly larger than that in human neurons (*Figure 4C*, p<0.01 two-way ANOVA with Holm-Sidak post hoc test). There were also small but significant differences in the $V_{0.5}$ of both inactivation (*Figure 4D*), and activation (*Figure 4E*), which were more hyperpolarized in human neurons (p<0.01). In contrast, recovery from inactivation (*Figure 4F*) was comparably rapid in neurons from both human and rat.

The kinetic properties of TTX-R currents in neurons from rat and human were also qualitatively similar (*Figure 5A and B*) with comparable rates of current activation (*Figure 5C*) and inactivation (*Figure 5D*).

## Pharmacological properties of TTX-R currents in rat and human DRG neurons

Two experiments were performed to enable comparison of the pharmacological properties of TTX-R currents in rat and human DRG neurons. In the first, the impact of the putatively NaV1.8 selective channel blocker, A-803467, was assessed. While this compound was shown to be both highly selective and potent against human NaV1.8 in heterologous expression systems (*Jarvis et al., 2007*), we were unable to obtain any evidence of an A-803467-induced decrease in TTX-R current in human DRG neurons at concentrations between 3 and 100 nM (n = 3–7 per concentration, data not shown). It was only at the very highest concentration tested (1 µM), that small but consistent block of TTX-R current was observed (*Figure 6A*). The same concentration of A-803467 blocked over 50% of TTX-R current in rat DRG neurons (*Figure 6B*). The difference in fractional block between human and rat obtained with this concentration was statistically significant (*Figure 6C*, p<0.01).

In the second pharmacological experiment, we assessed both resting and use-dependent block of TTX-R currents with lidocaine. The potency of lidocaine was comparable against resting TTX-R currents from rat and human DRG neurons (*Figure 7A,B and C*). However, not only did TTX-R currents from human DRG neurons demonstrate little use-dependent inactivation in the absence of lidocaine, there was no detectable use-dependent block of these currents in the presence of lidocaine (*Figure 7A and D*). This was in contrast to the use-dependent inactivation and block of TTX-R currents in rat DRG neurons in the absence and presence of lidocaine, respectively (*Figure 7B and D*, p<0.01 two-way ANOVA).

## Biophysical properties of TTX-S currents in rat and human DRG neurons

As with TTX-R currents, TTX-S currents in rat and human DRG neurons were qualitatively similar. There was also no significant difference between species with respect to the steady-state inactivation of TTX-S currents (*Figure 8A*). However, the voltage-dependence of TTX-S activation was significantly more hyperpolarized in human than in rat DRG neurons (*Figure 8B*, p<0.01). Furthermore, while over 80% of TTX-S current from human DRG neurons recovered from inactivation with a fast

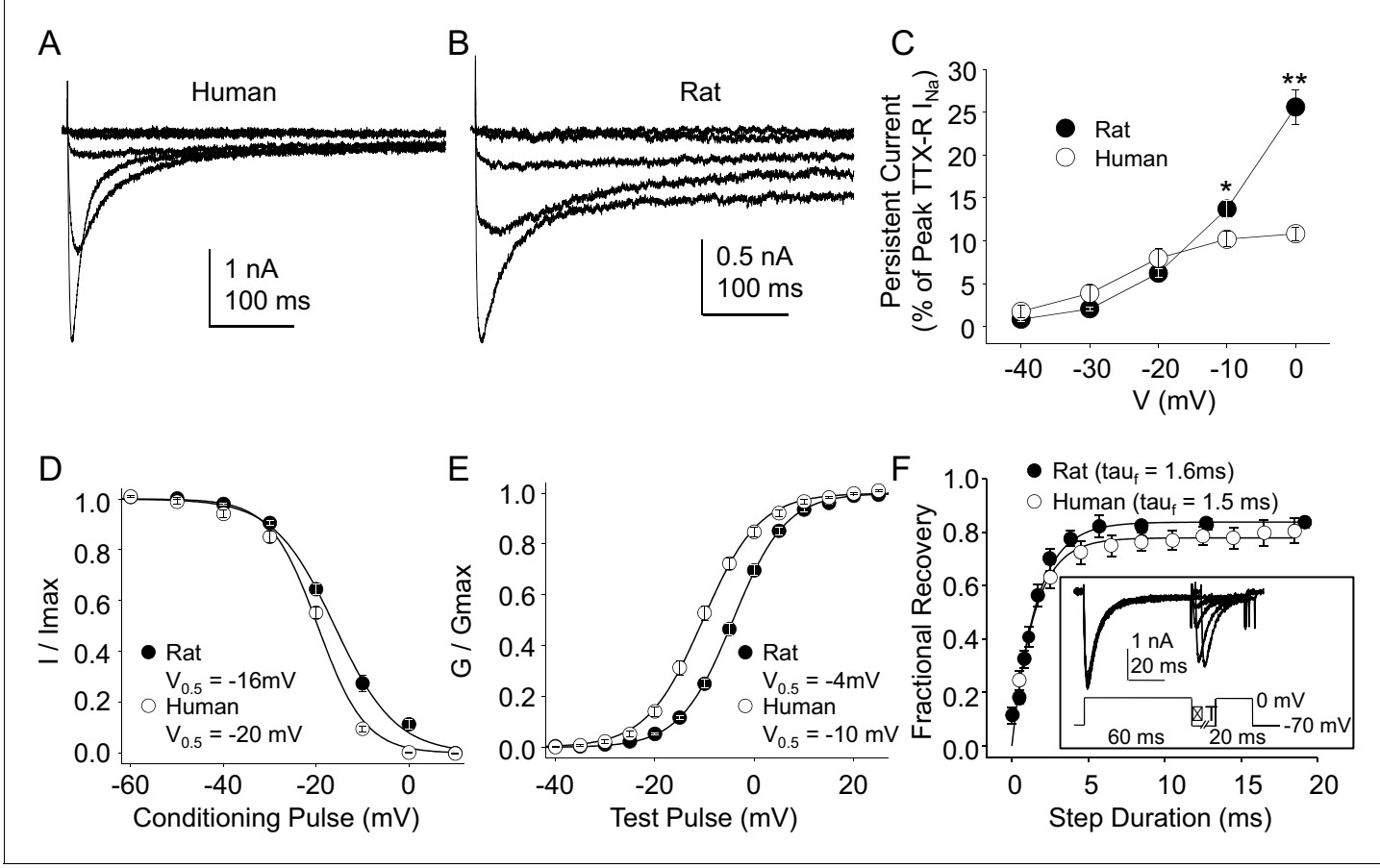

**Figure 4.** Steady-state biophysical properties of TTX-R currents in human and rat DRG neurons. (A) A 500 ms test pulse to potentials between −40 and 0 mV from a holding potential of −40 mV was used to assess the presence of persistent TTX-R current in human (A) and rat (B) DRG neurons. (C) The mean (± SEM) of persistent current analyzed as the % of the peak TTX-R current, is plotted relative to the voltage at which the current was evoked in neurons from rat (n = 30) and human (n = 34). (D) Steady-state inactivation of TTX-R current was assessed with the protocol shown in *Figure 1*. Availability curves were fitted with a modified Boltzmann equation to determine Imax, the slope and the voltage at which TTX-R current were half inactivated ($V_{0.5}$). Data for each neuron were then normalized to the calculated Imax. There was small but significant difference in the $V_{0.5}$ of inactivation of TTX-R current in neurons from rat (n = 25) and human (n = 99, although data plotted were 25 neurons to facilitate comparisons with rat data). (E) G-V curves were generated from I-V data and fitted with a modified Boltzmann equation to determine Gmax, the slope, and the voltage ($V_{0.5}$) at which conductance was half of Gmax. Data for each neuron were normalized to the calculated Gmax. There was a small but significant difference in the $V_{0.5}$ of activation of TTX-R current in neurons from rat (n = 25) and human (n = 123, although data plotted were again from 25 neurons to facilitate comparisons with rat data). (F) Recovery from inactivation of TTX-R current was assessed with a two pulse protocol shown at the inset, the extent of recovery from inactivation was determined by comparing the peak inward current evoked during the test pulse(second) to that evoked during the conditioning pulse(first). This ratio is plotted relative to the interpulse duration. Recovery curves were fitted with a double exponential. Recovery from inactivation of TTX-R current in neurons from rat (n = 10) and human (n = 13) were comparable.

The following source data is available for figure 4:

**Source data 1.** Data plotted in *Figure 4*.

time-constant, only 50% of TTX-S current from rat DRG neurons recovered as rapidly (*Figure 8C*, p<0.01).

TTX-S current kinetics in neurons from rat and human were also qualitatively similar (*Figure 9A* - D). However, current activation rate in human DRG neurons was faster than that in rat DRG neurons across the voltage range of current activation that was tested (*Figure 9C*).

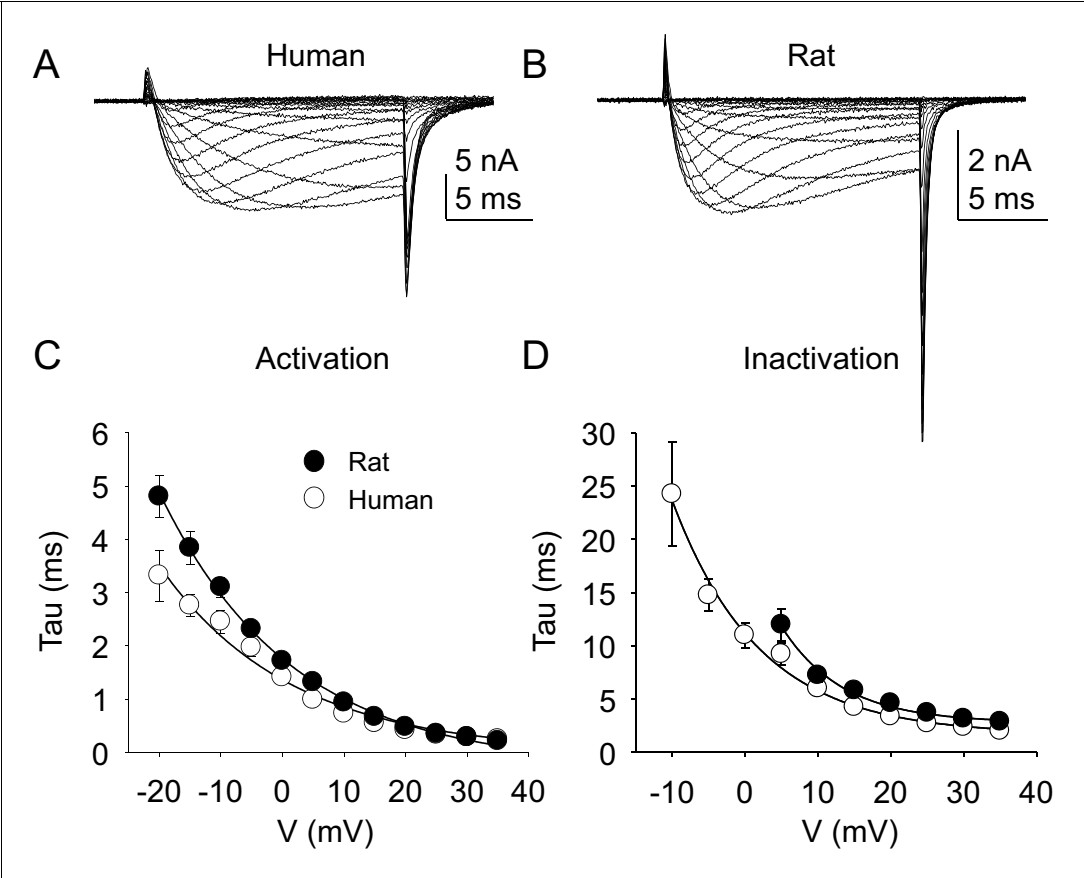

**Figure 5.** Kinetic properties of TTX-R current in human and rat DRG neurons. TTX-R I-V data from human (**A**) and rat (**B**) DRG neurons was used to assess the voltage-dependence of the rates of current activation (**C**) and inactivation (**D**). Currents were evoked with 15 ms voltage steps to potentials ranging between −40 mV and +40 mV, following a 500 ms pre-pulse to a potential at which TTX-S currents were completely inactivated and TTX-R currents were fully available for activation. That potential was −40 mV for the neurons studied in A, and B. The rising phase of TTX-R current at each potential was fitted with a single exponential to determine the rate of current activation. The falling phase of the TTX-R current during each depolarizing voltage step was fitted with a single exponential to determine the rate of current inactivation. Data in each plot are from 25 rat neurons and 27 human neurons.

The following source data is available for figure 5:

**Source data 1.** Data plotted in *Figure 5*.

## Pharmacological properties of TTX-S currents in rat and human DRG neurons

Four pharmacological experiments were also performed on TTX-S currents. The first was prompted by the initial observation that TTX-S currents in human DRG neurons were less sensitive to TTX than was our previous experience from rat DRG neurons. Consistent with this impression, there was a significant difference between TTX-S currents from rat and human DRG neurons with respect to the fraction of current blocked by 30 nM TTX. That is, only ~30% of the TTX-S current in human DRG neurons was blocked by this concentration of TTX (*Figure 10A*) in contrast to the >90% of current blocked in rat DRG neurons (*Figure 10B*). This difference was statistically significant (*Figure 10C*, $p<0.01$).

Lidocaine was again used in the second experiment. In contrast to the results obtained with TTX-R currents, there were differences between rat and human DRG neurons with respect to the resting block of TTX-S current, where lidocaine was less potent in the rat than the human (*Figure 11A,B and C*, $p<0.01$). Furthermore, while TTX-S currents in neurons from both rat and human were subject to use-dependent inhibition in the absence of lidocaine (*Figure 11A,B and D*),

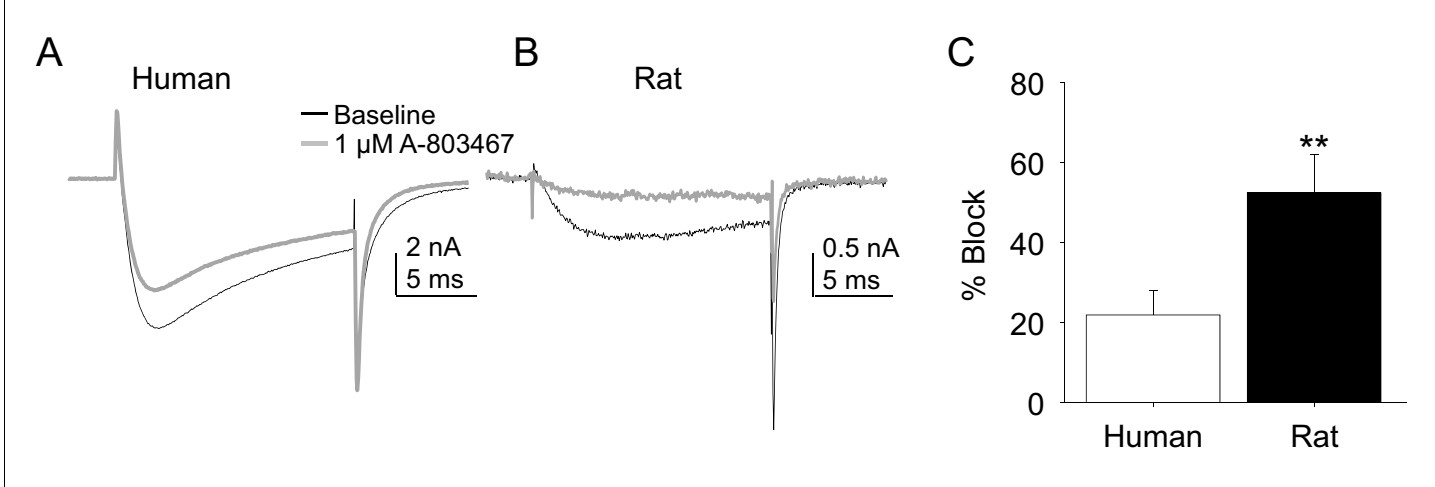

**Figure 6.** Impact of A-803467 on TTX-R current in human and rat DRG neurons. TTX-R current was evoked with a 15 ms depolarizing voltage step to 0 mV every 10 s, following a 500 ms prepulse to −65 mV in (**A**), and −40 mV in (**B**) Current evoked before (black) and after (grey) application of A-803467 (1 μM) to a human (**A**) and a rat (**B**) DRG neuron. Of note, as a prepulse to −65 mV would not inactivate all the TTX-S current in the human DRG neuron, the more rapid current activation likely reflects the contaminating presence of this faster current. Nevertheless, there is only a small reduction in current at the end of the test pulse. (**C**) The mean block of current in rat neurons (n = 4) was significantly greater than that in human neurons (n = 4). Of note, no detectable block was observed in eight other human neurons tested with lower concentrations of A-803467 (30–300 nM).

The following source data is available for figure 6:

**Source data 1.** Data plotted in *Figure 6*.

the inhibition was significantly (p<0.01) greater in rat neurons, at least at 1 and 2 Hz (*Figure 11D*). In both human and rat neurons, lidocaine caused significant (p<0.01) use-dependent block (*Figure 11A,B and D*).

The third and fourth experiments we designed to begin to assess the channel subtypes underlying the TTX-S current, given a growing body of evidence pointing to NaV1.7 as a therapeutic target for the treatment of pain (*Vetter et al., 2017*). As an initial foray into this question, we assessed the presence of a low threshold 'ramp current' in human DRG neurons based on rodent and heterologous expression data suggesting that because of the relatively slow development of closed-state inactivation of NaV1.7 channels, this subunit was responsible for large, low threshold currents evoked with a ramp depolarization. Despite the presence of TTX-S currents in every neuron included in this analysis, we detected no evidence of a low threshold ramp current in human DRG neurons in response to a depolarizing ramp from −100 mV to 0 mV over 500 ms, following a 500 ms pre-pulse to −90 mV (*Figure 12A*). Rather, peak inward current evoked in response to the ramp was at −5.9 ± 0.6 mV (n = 74). Because closed state-inactivation of NaV1.7 in human neurons may develop more rapidly than in rodent sensory neurons, we next assessed the impact of the NaV1.7 selective blocker Pro-Tx II (*Schmalhofer et al., 2008*) on TTX-S current in human DRG neurons. Little if any detectable suppression of TTX-S currents was observed in response to concentrations as high as 30 nM in the 27 neurons tested (*Figure 12B*). While we tested three different lots of the toxin on human neurons, we did not confirm the efficacy of any of these on rat DRG neurons. Thus, because of the notorious difficulty in working with large peptides, it is possible that negative results with this toxin were due to experimental errors. Nevertheless, we did test a fourth lot of the toxin, handled identically to the previous three, on rat DRG neurons, and observed a 50.5 ± 13.7% reduction in TTX-S current in the five neurons tested with 10 nM Pro-Tx II (*Figure 12B*).

While the ramp current and Pro-Tx II data argued against the presence of NaV1.7 in human DRG neurons, recent results with a putatively NaV1.7 selective small molecule blocker, PF-05089771 suggested that the majority of TTX-S current in human DRG neurons is carried by NaV1.7 (*Alexandrou et al., 2016*). Thus, in the fourth set of experiments, we assessed the impact of PF-05089771 on human DRG neurons. Consistent with previous results (*Alexandrou et al., 2016*), we

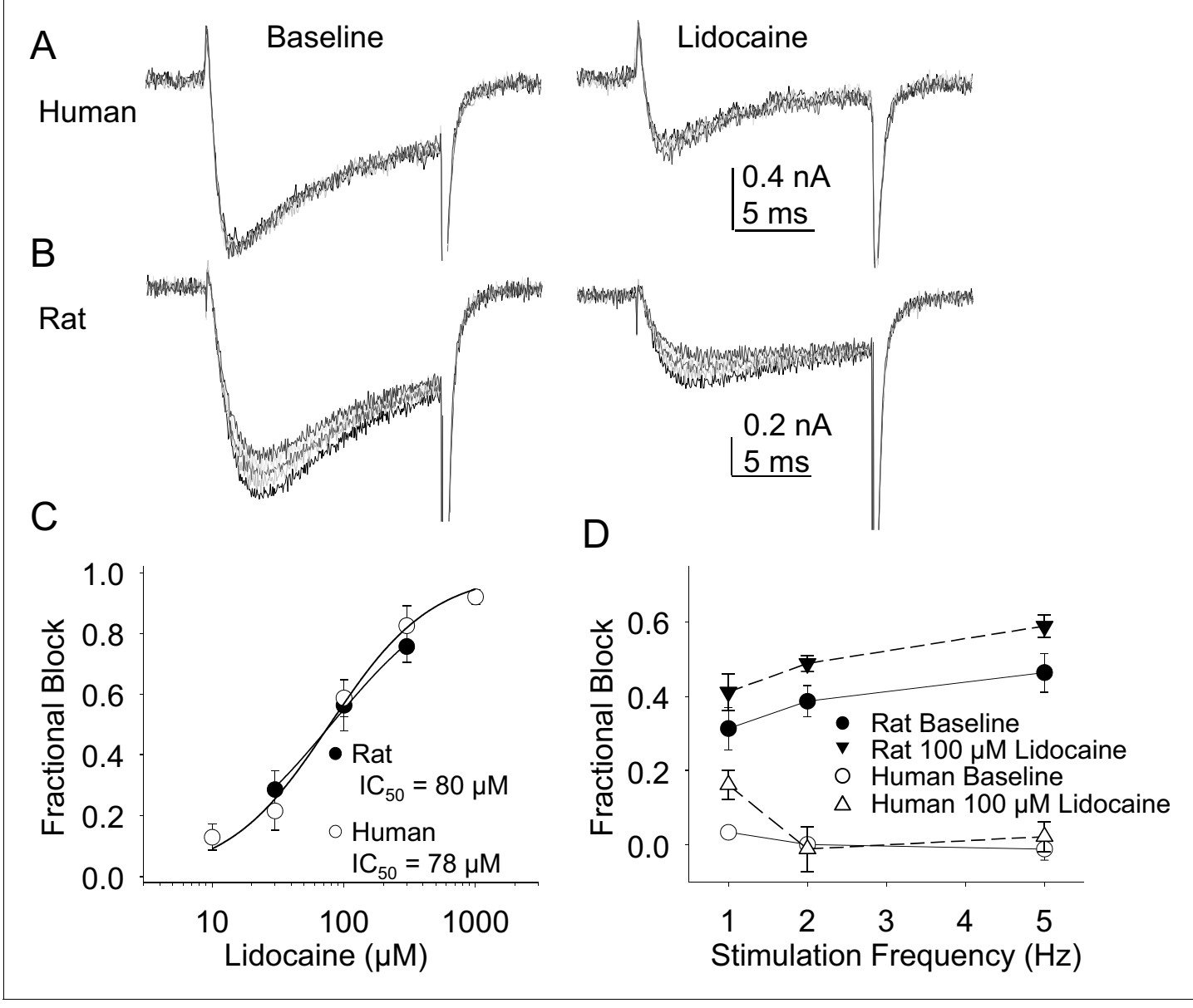

**Figure 7.** Resting and use-dependent block of TTX-R currents with lidocaine in human and rat DRG neurons. TTX-R current was evoked twenty times with a voltage step to 0 mV at 1, 2 and 5 Hz before and after the application of lidocaine. The responses to the first five voltage steps evoked at 2 Hz before and after 100 µM lidocaine are shown for a human (A, evoked from a holding potential of −40 mV) and rat (B, evoked from a holding potential of −35 mV) DRG neuron. (C) Lidocaine-induced steady-state block of TTX-R current was assessed ~three minutes after application of each concentration of lidocaine, prior to the initiation of the use-dependent block protocols. The steady-state block of currents in human (n = 9) and rat (n = 10) neurons were comparable. (D) Use-dependent block, calculated as the fraction of current evoked at the 20th pulse relative to that evoked with the first (P20/P1). Use-dependent block of TTX-R current observed in rat DRG neurons was increased in the presence of lidocaine. In contrast, there was little use-dependent block of TTX-R currents in human DRG neurons in the absence of lidocaine, and the only evidence of a lidocaine-induced increase in use-dependent block was observed at a stimulation frequency of 1 Hz.

The following source data is available for figure 7:

**Source data 1.** Data plotted in *Figure 7*.

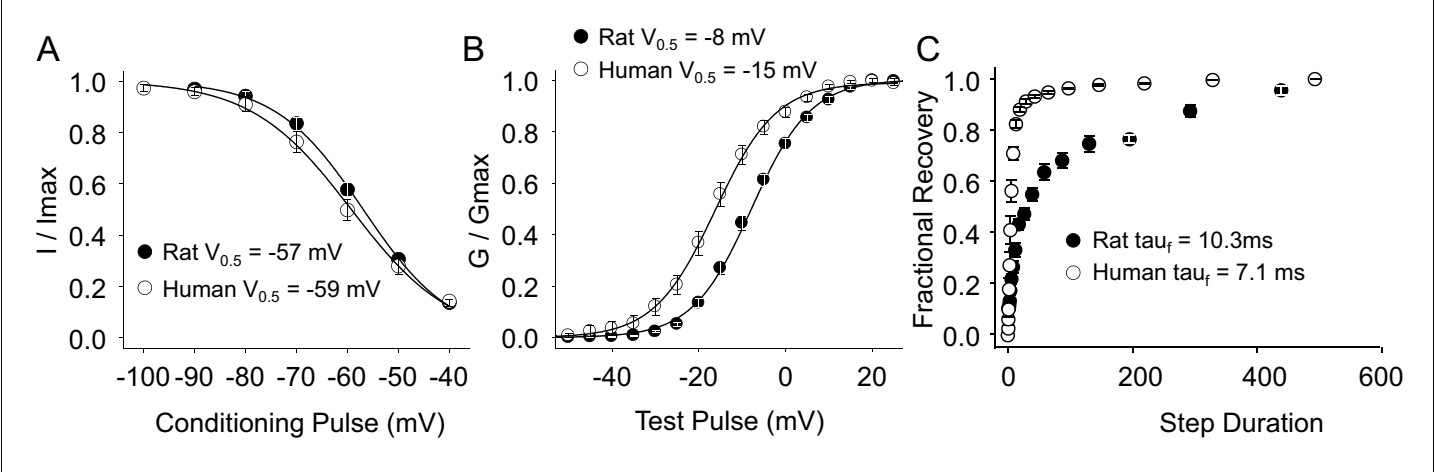

**Figure 8.** Steady-state biophysical properties of TTX-S current in human and rat DRG neurons. TTX-S currents were isolated as described in *Figure 1*. Steady-state availability (**A**), activation (**B**), and recovery from inactivation (**C**) data were collected for TTX-S currents as described for TTX-R currents in *Figure 4*, except that the holding and recovery potential were −90 mV. Availability data were from 29 rat neurons and 99 human neurons (although data from only 29 neurons are plotted, to facilitate comparisons between human and rat). G-V data were also from 29 rat neurons, and from 128 human neurons (although data from only 29 neurons is plotted). Recovery data are from the 8seven rat neurons and 8 human neurons. The faction of current recovered with a fast time constant in human neurons was significantly greater than that in rat neurons.

The following source data is available for figure 8:

**Source data 1.** Data plotted in *Figure 8*.

observed complete inhibition of TTX-S currents in human DRG neurons following a 30 min pre-incubation with PF-05089771 with concentrations as low as 30 nM (*Figure 12C and D*). Concentration-response data indicated that the IC50 for TTX-S current block was ~6 nM, close to that previously reported (*Alexandrou et al., 2016*). However, in contrast to the previous study of PF-05089771 on human DRG neurons, which was performed in the presence of A-803467, we assessed the impact of this compound on TTX-R currents. In striking contrast to the heterologous expression data indicating NaV1.8 is resistant to PF-05089771 at concentrations as high as 10 μM, we observed a significant reduction in TTX-R currents in neurons treated with this compound. In fact, the calculated IC50 for inhibition of TTX-R currents in human DRG neurons was ~50 nM (*Figure 12C and D*).

## Discussion

The purpose of the present study was to compare the properties of voltage-gated Na$^+$ currents in DRG neurons from rat and human. As with currents in the rat, the major types of current, those classically described as the rapidly activating and rapidly inactivating TTX-S current, and the more slowly activating and slowly inactivating TTX-R current, were well isolated with voltage clamp protocols. In fact, an issue potentially relevant to the interpretation of our pharmacological experiments, voltage protocols were used to isolate these currents in all but our initial characterization experiments. Both types of current were potentiated by inflammatory mediators, although TTX-S currents were increased in only a subpopulation of neurons in which TTX-R currents were increased. The biophysical properties of TTX-R currents in human DRG neurons were qualitatively similar to those in rat DRG neurons, with small, but significant differences in the $V_{0.5}$ of current inactivation and activation. However, in contrast to TTX-R current in rat DRG neurons, there was little evidence of use-dependent inactivation of currents in human DRG neurons. There were also significant differences in the extent of the block produced by the putatively NaV1.8 selective blocker A-803467, as well as the use-dependent block produced by lidocaine, both of which were significantly smaller on currents from human DRG neurons. As with TTX-R currents, TTX-S currents in rat and human DRG neurons were qualitatively similar. However, the $V_{0.5}$ of current activation was more hyperpolarized, a larger fraction of current recovered rapidly from inactivation, and the currents activated significantly faster

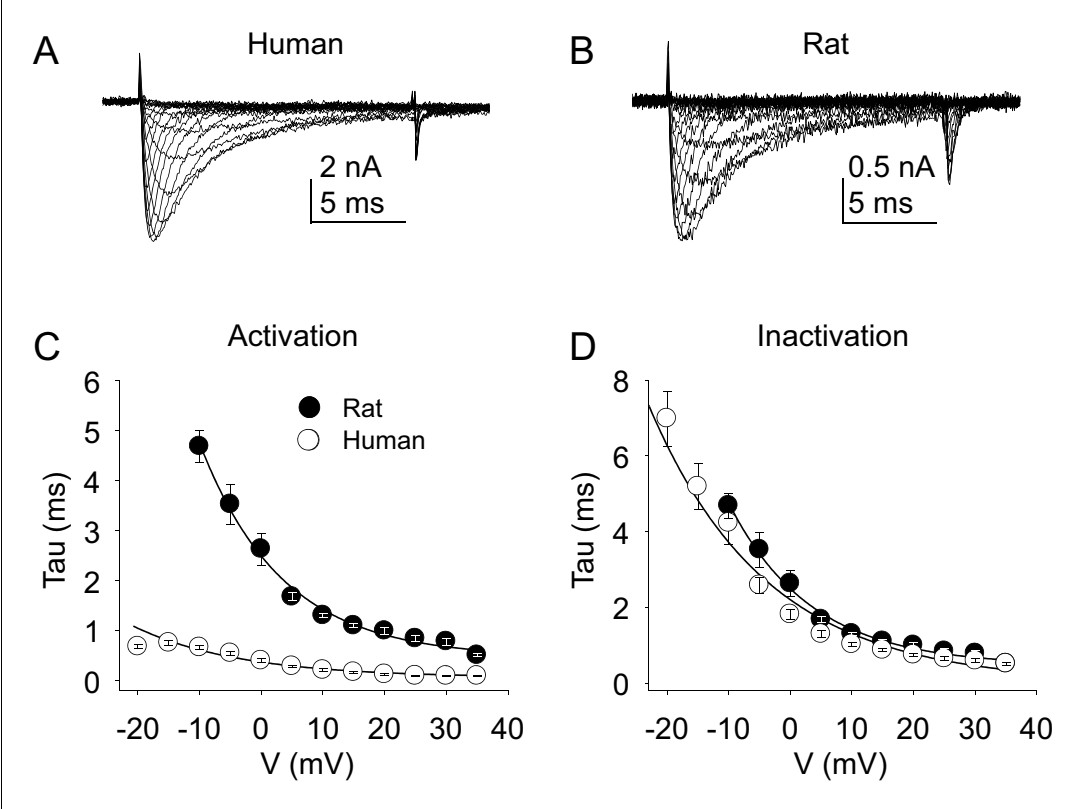

**Figure 9.** Kinetic properties of TTX-S current in human and rat DRG neurons. TTX-S I-V data from human (**A**) and rat (**B**) DRG neurons was used to assess the voltage-dependence of the rates of current activation (**C**) and inactivation (**D**). Current was isolated as described in **Figure 1**, where in the neurons shown in (**A**), and (**B**), the pre-pulse potential was −90 mV to evoke total current, and –35 and –40 mV to inactivate TTX-S currents in the human and rat neuron, respectively. Activation and inactivation rates were determined as described in **Figure 5**. The activation of currents in human DRG neurons (n = 128, although data from only 29 are plotted to facilitate comparisons with rat data) was significantly faster, and demonstrated significantly less voltage-dependence than TTX-S currents from rat DRG neurons (n = 29). However, inactivation rates and the voltage-dependence of this process was comparable in currents from rat and human neurons.

The following source data is available for figure 9:

**Source data 1.** Data plotted in **Figure 9**.

in human DRG neurons. The potency of TTX was significantly lower against currents in human DRG neurons. The lidocaine-induced use-dependent block of TTX-S currents was also smaller in human DRG neurons. Finally, there was no evidence of low threshold ramp currents in human DRG neurons, the TTX-S currents were resistant to the NaV1.7-selective blocker Pro-Tx II, and while TTX-S currents were blocked by PF-05089771, this putatively selective small molecule inhibitor of NaV1.7 also blocked TTX-R currents in human DRG neurons. These results have important implications for drug development, as well as our understanding of pain mechanisms.

With respect to drug development, the results of the present study are consistent with those from previous studies indicating that both expression systems and species differences may have a significant influence on the pharmacological properties of the protein in question. This may be particularly true for VGSC because of extensive post-translational modifications (*Laedermann et al., 2015*). Potentially more problematic is that these modifications may not only be cell type specific, but specific to location(s) within a given cell (*Harriott and Gold, 2008*). In the context of pharmacology, where side effects are often due to off-target actions of a drug, any decrease in relative potency increases the likelihood of side effects. We observed a loss of selectivity in human DRG neurons for both A-803467 and PF-05089771. In the case of the former, we observed a decrease in TTX-S current in human DRG neurons in response to 1 µM A-803467 which was close to that of

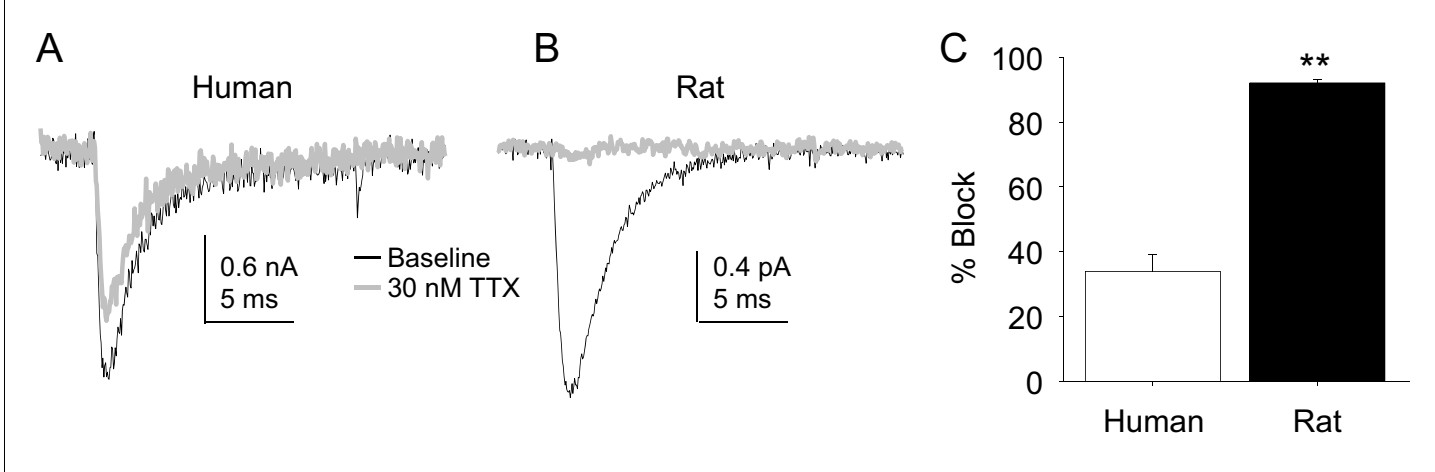

**Figure 10.** The potency of TTX block of TTX-S currents was lower in human than in rat DRG neurons. TTX-S currents isolated as described in *Figure 1* where in the neurons shown in (**A**) (human), and (**B**) (rat), the pre-pulse potential was −90 mV to evoke total current, and −40 mV to inactivate TTX-S currents in both the human and rat neurons shown. Currents were evoked before and after 30 nM TTX application. (**C**). The fractional block of TTX-S current produced by 30 nM was significantly (p<0.01) greater in rat (n = 9) than in human (n = 8) DRG neurons.

The following source data is available for figure 10:

**Source data 1.** Data plotted in *Figure 10*.

TTX-R currents (data not shown), while in the case of the latter, we observed a block of TTX-R currents with an IC50 less than an order of magnitude higher than that for block of TTX-S currents. These results suggest that despite the very high degree of selectivity observed with these compounds on heterologously expressed VGSC, the therapeutic window for A-803467 would likely have been quite limited in the clinical setting, while any therapeutic effect of PF-05089771 may also reflect block of NaV1.8 channels. Similarly, the demonstration in heterologous expression systems that the potency of VGSC blockers may be state-dependent served as the rationale for the development of state-dependent VGSC blockers for the treatment of pain (*Dick et al., 2007*). That is, this observation suggested it may be possible to use such drugs to preferentially block spontaneous or aberrant activity, and thereby reduce spontaneous pain while preserving normal sensation. However, our observation that TTX-R currents in human DRG neurons, currents thought to play a critical role in spike initiation in nociceptive afferents, are subject to little if any use-dependent block in either the absence or presence of a prototypical use-dependent VGSC blocker could explain the limited therapeutic efficacy of local anesthetics (at least when delivered at concentrations below that needed to block action potential propagation) in pain patients (*Finnerup et al., 2015*).

There are several implications of the results of the present study with respect to our understanding of pain mechanisms. First, consistent with results from another recent study of human DRG neurons, our data confirm the utility of this model for the analysis and/or confirmation of second messenger systems implicated in the sensitization of putative nociceptive afferents. That is, it was recently demonstrated that application of inflammatory mediators to isolated human sensory neurons resulted in an increase in excitability (*Davidson et al., 2014*). Furthermore, activation of group II metabotropic glutamate receptors, shown to block the inflammatory mediator-induced sensitization of mouse sensory neurons, was also shown to block the sensitization of human sensory neurons (*Davidson et al., 2016*). Results of the present study confirm previous results from the study of rodent sensory neurons, suggesting that increases in both TTX-R (*Gold et al., 1996*) and TTX-S (*Cardenas et al., 1997*) currents are likely to contribute to this increase in excitability. The observation that TTX-S currents were only increased in a subpopulation of neurons in which TTX-R currents were increased, is interesting for at least two reasons. One is that it suggests that there is relatively tight coupling between G-proteins and effector molecules, even within the isolated sensory neurons. This would underscore the importance of getting a therapeutic intervention to the right place.

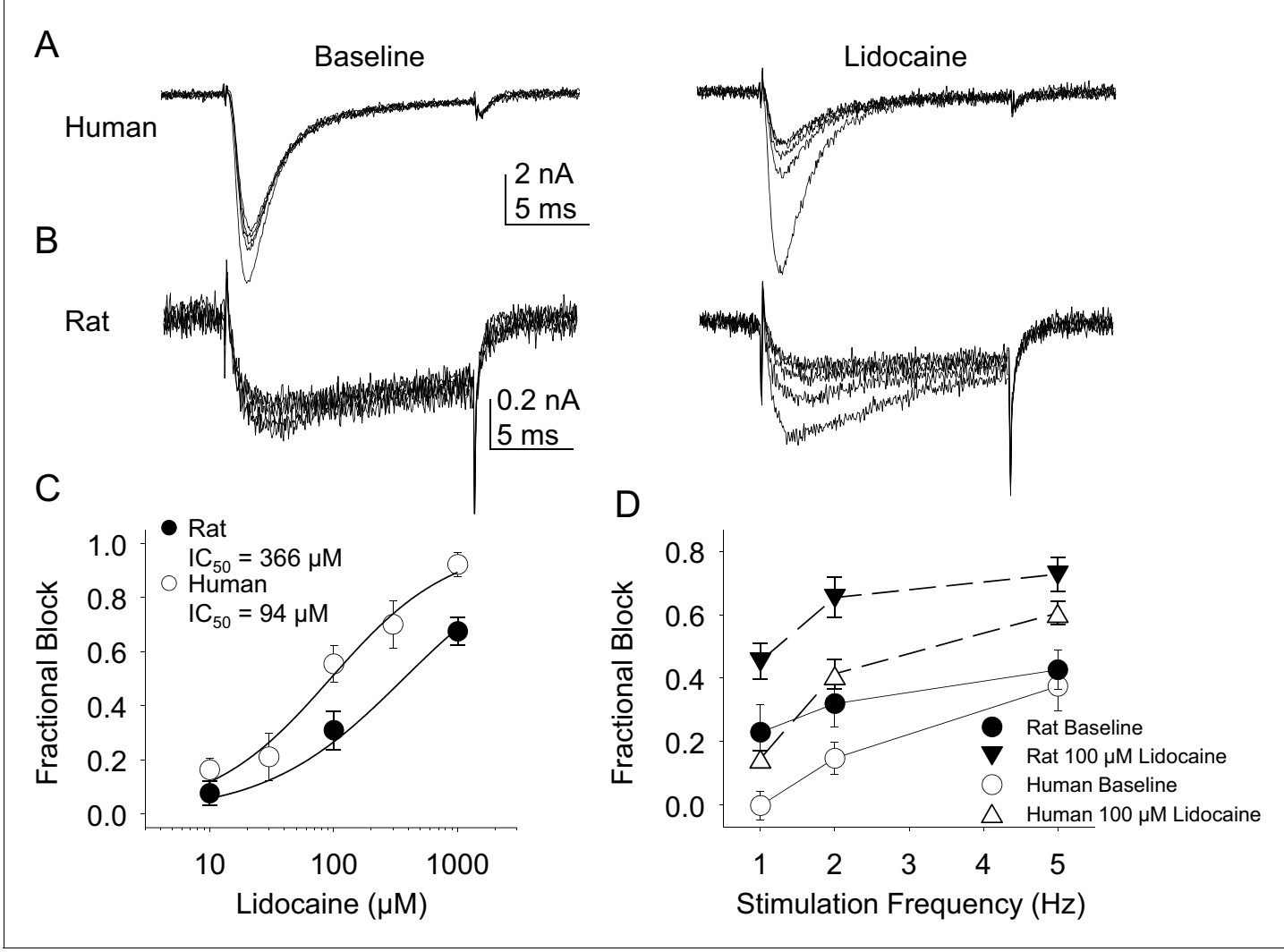

**Figure 11.** Resting and use-dependent block of TTX-S currents with lidocaine in human and rat DRG neurons. Currents were evoked in neurons from human (**A**) and rat (**B**), as described in *Figure 7* with a voltage step to −25 mV from a holding potential of −90 mV. (**C**). Lidocaine-induced steady-state block of TTX-S currents was also determined as in *Figure 7*. However, in contrast to TTX-R currents, the potency of lidocaine-induced block of TTX-S currents in human DRG neurons (n = 8) was significantly higher than that in rat DRG neurons (n = 13). (**D**). Lidocaine was associated with a significant increase in use-dependent block of TTX-S currents, assessed at 1, 2 and 5 Hz, in both human and rat neurons.

The following source data is available for figure 11:

**Source data 1.** Data plotted in *Figure 11*.

Another reason the observation about the modulation of TTX-S and R currents is interesting is that it suggests that it may be necessary to reduce both TTX-R and TTX-S currents to achieve maximal pain relief.

Another implication of our results concerns assumptions about the relationship between cell body diameter and afferent function. That is, there is compelling evidence to suggest that at least for cutaneous afferents, nociceptors are enriched in the subpopulation with a small cell body diameter while low-threshold afferents are enriched in a subpopulation of neurons with a large cell body diameter (*Lawson, 2002*; *Djouhri et al., 2003*). Consistent with association between cell body size and function, NaV1.8 is not only enriched in nociceptive afferents (*Djouhri et al., 2003*), but is preferentially expressed in sensory neurons with a small cell body diameter. In contrast, neither the size distribution nor the distribution of TTX-R currents among human DRG neurons suggests that there is a

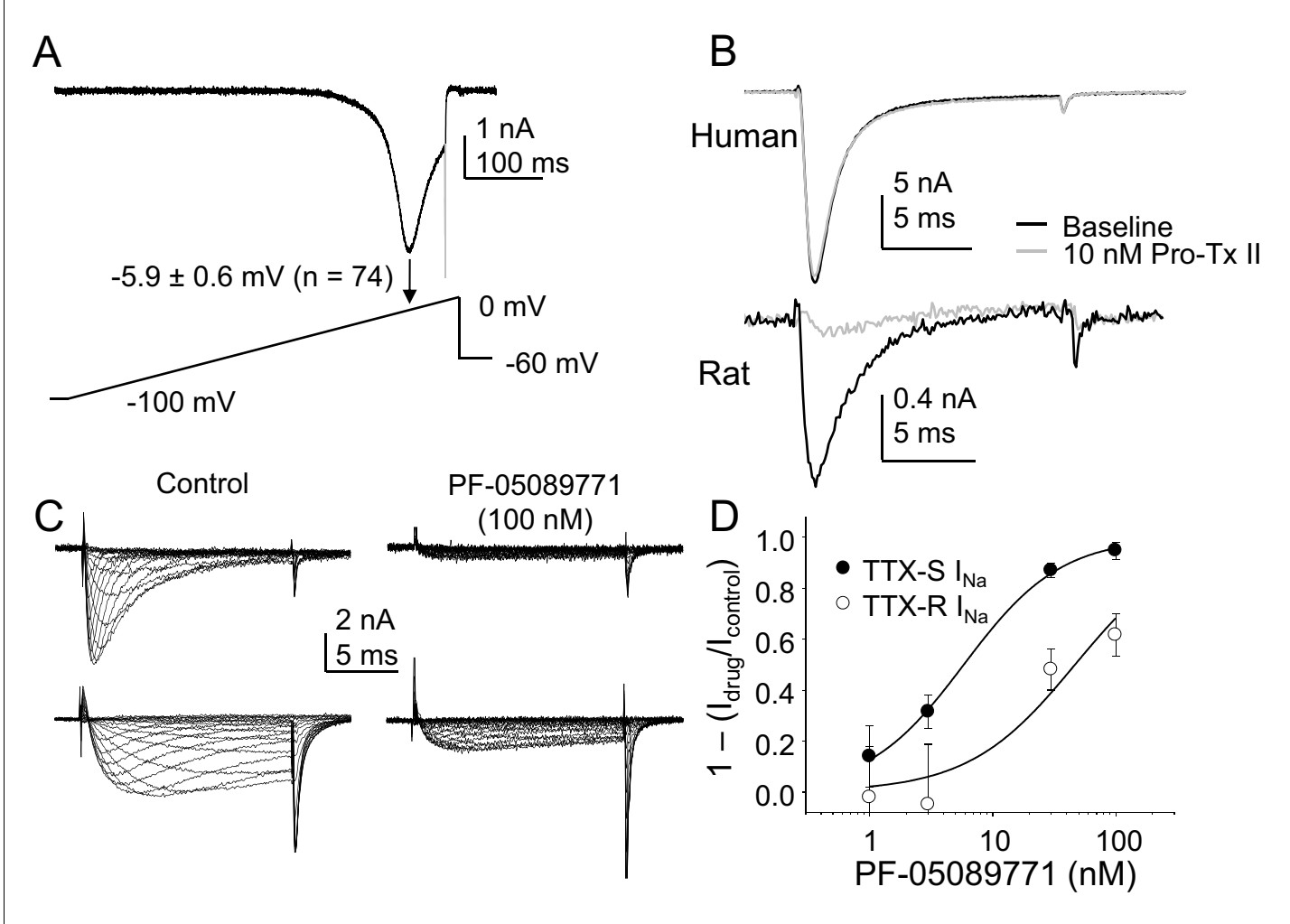

**Figure 12.** The contribution of NaV1.7 to TTX-S currents in human DRG neurons. (A) Ramp currents were evoked with a voltage-clamp protocol consisting of a depolarization from −100 mV to 0 mV over 500 ms following a 500 ms voltage-step to −90 mV. Peak inward current evoked in response to the ramp was seen at −5.9 ± 0.6 mV (n = 74). (B) Top traces: TTX-S currents evoked at 0 mV, isolated as described in *Figure 1* from a human DRG neuron with 500 ms voltage-steps to −90 mV and −35 mV prior to the voltage-step to 0 mV, to activate total current and inactivate TTX-S current, respectively. Currents were evoked before and after 10 nM Pro-Tx II. Comparable data were obtained in 8 other neurons tested. Bottom traces: The same concentration of toxin blocked a fraction of TTX-S currents in a rat DRG neuron, isolated with 500 ms voltage-steps to −90 mV and −40 mV prior to the voltage-step to −5 mV. While the fraction of block was variable, a fraction of TTX-S current was blocked in the five other neurons studied. (C) Pre-incubating human DRG neurons with 100 nM PF-05089771 for 30 min prior to study (left traces) was associated with an almost complete block of TTX-S currents (top traces), as well as a significant reduction in TTX-R currents (bottom traces). TTX-S current isolated from TTX-R current as described in *Figure 1*, with pre-pulse potential to −90 mV and −35 mV, for both the control neuron (left traces) and the treated neuron (right traces). Control neurons were run in parallel with pre-incubation times in bath solution identical to the neurons pre-incubated with PF-05089771. (D) The inhibition of both TTX-S and TTX-R currents by PF-05089771 was concentration dependent, were the average current density in control neurons run in parallel with treated neurons was used to assess the magnitude of current block. Pooled data are from three donor, with 5–6 neurons per concentration and 13 neurons in the control group. Data were fitted with a modified Hill equation, which yielded an IC50 of 6 nM for block of TTX-S current and 47 nM for block of TTX-R current.

The following source data is available for figure 12:

**Source data 1.** Data plotted in *Figure 12*.

comparable relationship between cell body size and function in the human as has been documented in the rodent. Given that we did not assess the currents present in the very largest human DRG neurons, it is possible that we missed the subpopulation of putative non-nociceptive afferents comparable to that in the rodent, but this would still suggest TTX-R currents are far more widely distributed in human neurons than in the rodent. In this regard, it is also possible that TTX-R currents play a more important role in non-nociceptive human than rodent neurons. However, our inflammatory mediator data, as well as previous excitability data from human DRG neurons (*Davidson et al., 2014*) suggests the association between TTX-R currents and the nociceptive phenotype is preserved in human DRG neurons. Minimally, these data suggest that additional criteria will be needed to identify putative subtypes of human sensory neurons in future studies.

A third implication of our results concerns the present model of spike initiation and the emergence of sustained activity following injury in putative nociceptive neurons. That is, data from the study of rodent sensory neurons suggests that spike initiation involves the initial activation of NaV1.7 followed by the activation of NaV1.8 (*Cummins et al., 1998*). This suggested NaV1.7 as a viable target to attenuate nociceptor activity. However, our data from human DRG neurons suggest that while the threshold for activation of TTX-S currents is more negative than that of TTX-R currents, the failure of TTX to block ramp currents in the nine neurons tested (data not shown) suggests that ramp currents are dominated by TTX-R currents. This suggests that TTX-R currents are responsible for both spike initiation and the action potential over-shoot in human DRG neurons. Similarly, while use-dependent block of TTX-R currents in rodent sensory neurons has been argued to contribute to the slow adaptation observed in nociceptive afferent in response to prolonged stimulation (*Choi et al., 2007*), the absence of use-dependent block of TTX-R currents in human DRG neurons would enable these channels to underlie sustained neural activity such as that associated with ongoing pain. Furthermore, the relatively fast and complete recovery from inactivation observed for human TTX-S currents suggests that even in the absence of injury, these currents would be able to contribute to a relatively high level of sustained activity. This is in contrast to current models of injury-induced changes in VGSC expression, where the upregulation of NaV1.3, a channel with a relatively rapid rate of recovery from inactivation (*Cummins and Waxman, 1997*), is thought to contribute to the increase in sensory neuron excitability observed following peripheral nerve injury.

Our data on the channel subunits underlying the TTX-S current in human DRG neurons also has important implications given the recent focus on NaV1.7 as a potential therapeutic target for the treatment of pain based on the human channelopathy data (*Dib-Hajj et al., 2013*). One interpretation of our PF-05089771 data is that NaV1.7 underlies most, if not all of the TTX-S current in human DRG neurons. This interpretation would be consistent with that proposed by Alexandrou and colleagues based on the selectivity of this compound on human Na$^+$ channel subtypes expressed in HEK293 cells, as well as the impact of this compound on TTX-S current in human DRG neurons (*Alexandrou et al., 2016*). If this interpretation was correct, however, it would suggest that the biophysical properties of NaV1.7 in human DRG neurons, at least with respect to the relatively slow entry into an inactivated state, are different from that of human NaV1.7 channels expressed in HEK293 cells, or NaV1.7 channels in rodent sensory neurons (*Cummins et al., 1998*). That is, because closed state inactivation develops so much more slowly in heterologously expressed NaV1.7 channels than other subtypes such as NaV1.4, NaV1.7 was proposed to account for the large low threshold TTX-S current evoked with ramp depolarization of small rodent DRG neurons (*Cummins et al., 1998*). However, only high threshold ramp currents were evoked in human DRG neurons that were resistant to TTX. Furthermore, the interpretation that NaV1.7 is the dominant channel subtype underlying TTX-S currents in human DRG neurons would also suggest that the potency of Pro-Tx II is different for the block of NaV1.7 in human DRG neurons than in HEK 293 cells (*Schmalhofer et al., 2008*) or rodent sensory neurons (*Laedermann et al., 2014*).

An alternative interpretation of our results is that NaV1.7 may not be the dominant subunit underlying TTX-S currents, let alone significantly contribute to these currents in human DRG neurons. That is, while it is possible that PF-05089771 retains its selectivity for NaV1.7 over other natively expressed TTX-S channels, our observation that this compound blocks TTX-R currents in human DRG neurons with an IC50 of ~50 nM, yet has no activity at NaV1.8 channels in HEK293 cells at concentrations as high as 10 µM (*Alexandrou et al., 2016*) raises the possibility that there is a more generalized loss of selectivity of this compound against Na$^+$ channels present in their native environment. Importantly, while the channel block produced by PF-05089771 was shown to be highly state-

dependent in heterologous expression experiments, blocking inactivated channels with a potency almost four orders of magnitude higher than that of channels in a resting or closed state (*Alexandrou et al., 2016*), this property is unlikely to account for the apparent block of TTX-R currents observed in human DRG neurons. That is, the pre-pulse potential used to evoke TTX-R currents would have left the majority of channels underlying the TTX-R current in a closed or resting state (i. e., fully available for activation), and therefore should have decreased, rather than increased the potency of this compound. A relative dearth of NaV1.7 in human DRG neurons would also account for the absence of a low threshold TTX-S ramp current, as well as the absence of channel block with ProTx-II. A limited expression of NaV1.7 in a subpopulation of human DRG neurons would also explain why the inflammatory mediator-induced increase in TTX-S current was only detected in a subpopulation of neurons. It is important to point out that even in rodent sensory neurons, other TTX-S channel subunits are expressed at levels comparable to that of NaV1.7, at least in subpopulations of neurons. For example, RNAseq analysis of mouse DRG neurons suggests that the number of copies of NaV1.6 (31.1) is comparable to that of NaV1.7 (54), at least in the subpopulation of neurons not in the TRPV1 lineage (*Goswami et al., 2014*). Comparable results were obtained with a quantitative PCR analysis of single DRG neurons, where in those larger than 30 µm in diameter, NaV1.6 expression was comparable to that of NaV1.7 (*Ho and O'Leary, 2011*). It is also worth noting that despite evidence that NaV1.7 is distributed throughout sensory neurons (*Black et al., 2012*), recent evidence suggests that the contribution of NaV1.7 to even rodent sensory neurons may have been overestimated. That is, while the activity of the putatively NaV1.7 selective spider venom peptide Pn3a on native TTX-R currents was not well described, results with this peptide suggests that NaV1.7 does not even account for 50% of the TTX-S current in the majority of rat DRG neurons (*Deuis et al., 2017*). It should also be noted that recent data from the NaV1.7 null mutant mice suggest that this subunit may, in fact play a more important role in mediating $Na^+$ influx and associated changes in gene expression than it does in the electrical properties of these neurons (*Minett et al., 2015*). And while compensation may account for the relatively limited impact of the NaV1.7 knock-out on TTX-S currents in mouse small diameter DRG neurons, the ~25% decrease in TTX-S current density observed in these neurons is consistent with a relatively limited contribution of NaV1.7 to the TTX-S current (*Nassar et al., 2004*). More relevantly, the suggestion that NaV1.7 contributes little to the TTX-S current in human sensory neurons is consistent with the relatively limited impact of gain of function mutations on the excitability of human nociceptive afferents (*Namer et al., 2015*).

While several potentially important differences were observed in the $Na^+$ currents in human and rat DRG neurons, it is important to consider the extent to which these differences were due to experimental variables rather than species differences. Neurons were collected from both male and female human donors with an average age of 45 years, at least an hour after cross clamp. In contrast, neurons were obtained from relatively young (~60 day old) male rats that were deeply anesthetized at the time of tissue collection. Unfortunately, we were unable to rule out the time between cross-clamp and tissue collection as a factor contributing to the differences observed, as preliminary results indicated that the viability of rat DRG neurons fell off precipitously 15 min after death, even in rats perfused with ice-cold saline. However, we suggest that neither of the other two main factors were likely to contribute to the differences observed, as there was no detectable influence of either age (at least comparing neurons from young (<30) and old (>65) donors) or donor sex on the biophysical properties of $Na^+$ currents assessed (data not shown). It is also possible that the larger currents and larger cell body size of human DRG neurons resulted in a decrease in clamp control that contributed to the hyperpolarizing shift in both TTX-R and TTX-S current G-V curves as well as the faster rates of current activation observed in human neurons. Arguing against this possibility, however, is the fact that while the slopes of the G-V curves for both current types were steeper in human than in rat neurons ($4.8 \pm 0.1$ mV and $5.4 \pm 0.2$ mV vs $5.0 \pm 0.1$ mV and $6.3 \pm 0.1$ mV, for TTX-R and TTX-S currents in human and rat neurons, respectively), these small differences could not account for the observed differences in the $V_{0.5}$ of activation, nor are they consistent with a loss of clamp control, which would be predicted to result in a much larger decrease in the slope of the G-V curve. In addition, differences in the $V_{0.5}$ of activation persisted when data from only the smallest human and largest rat neurons were compared. For example, the $V_{0.5}$ of current activation for TTX-R in the overlapping subpopulation of rat and human sensory neurons were $-2.1 \pm 0.4$ mV and $-10.8 \pm 1.7$ mV, respectively.

We also suggest that it is unlikely that the recording conditions used, which were necessary to both minimize voltage-gated $Ca^{2+}$ currents while maximizing patch stability had a significant influence on the biophysical properties of the currents recorded. That is, the properties of the currents observed in rat neurons recorded under conditions identical to those used for the study of human neurons, were comparable to those we have previously observed (*Gold et al., 2003*; *Flake et al., 2004*; *Vaughn and Gold, 2010*). That said, we have always used a relatively low concentration of extracellular $Na^+$ to help maintain clamp control, and minimized Ca2+ current contamination with a reduction of extracellular Ca2+ and/or the use of extracellular Ca2+ channel blockers such as Cd2+. In contrast, others, who have reported considerably more hyperpolarized potentials for the activation and inactivation of both TTX-S and TTX-R currents in rat DRG neurons have used higher concentrations of extracellular $Na^+$ as well as intracellular solutions containing fluoride (*Cummins and Waxman, 1997*; *Roy and Narahashi, 1992*; *Rush and Elliott, 1997*). It remains to be determined whether recording conditions have comparable influences on the biophysical properties of $Na^+$ currents in human DRG neurons. Finally, it is important to point out that while we attempted to control for the impact of state-dependent properties of the compounds tested, our pharmacological results should be interpreted with caution because of the potentially confounding interaction between these properties and the voltage protocols used to isolate TTX-S from TTX-R currents.

In summary, we have described important similarities and differences between human and rat dorsal root ganglion neurons with respect to the biophysical and pharmacological properties of the two major classes of voltage-gated $Na^+$ current. We would argue that the similarities were sufficient to justify the continued use of rodent sensory neurons as a model system with which to explore the potential contribution of a variety of channels critical to both pain and analgesia. However, we would also argue that the differences were sufficient to contribute to what continues to be a tremendously high failure rate in the development of novel analgesics. Use of human DRG neurons for the screening of potentially novel therapeutics targeting the primary afferent and/or proteins in the primary afferent may help rectify this situation, by enabling the identification of compounds unlikely to achieve the intended effect prior to the initiation of expensive clinical trials.

## Materials and methods

### Human tissue

L4 and L5 DRG were collected from organ donors with the consent of family members for the use of their loved one's tissue for research purposes. The protocol for the collection and study of tissue from organ donors was approved by the University of Pittsburgh Committee for Oversight of Research and Clinical Training Involving Decedents. As previous data on the biophysical and pharmacological properties of $Na^+$ currents in human DRG neurons was extremely limited, no a priori analyses were performed to estimate the number of donors needed to complete this study. Rather, we studied tissue from each donor entered into the study with the same basic protocol needed to extract basic features of $Na^+$ current types from each neuron, and then added additional analyses with the goal of obtaining data for each endpoint with neurons from at least three donors. Post-hoc analysis was then performed to rule out an influence of sex or age, the only major demographic features sufficiently represented in our data-set to perform such an analysis, on any of the biophysical properties determined for the majority of neurons from all donors.

### Rat tissue

Adult (250–320 g) male Sprague-Dawley rats (Envigo, Indianapolis, IN)) were used for all experiments. Rats were housed two per cage in a temperature and humidity controlled, Association for Assessment and Accreditation of Laboratory Animal Care International (AAALAC) accredited animal housing facility on a 12 hr:12 hr light:dark schedule. Food and water were available ad libitum. All procedures were approved by the University of Pittsburgh Institutional Animal Care and Use Committee and performed in accordance with National Institutes of Health guidelines for the use of laboratory animals in research. The number of rats used in this study was based on estimates made from variability observed in our previous biophysical and pharmacological analysis of $Na^+$ currents in rat sensory neurons as well as previous experience with both the amount of time needed to complete each protocol and consequently the number of neurons that could be studied from each rat in the

time window in which neurons were considered 'acutely' dissociated. As our previous data indicates that there is considerably more heterogeneity between neurons within the same rat that between rats, the use of three rats for any given endpoint enables us to detect any potential issues associated with any single preparation of neurons. Thus, neurons from at least three rats were used for each endpoint.

## Isolation and plating of human sensory neurons

DRG were obtained from organ donors following collection of tissue needed for transplantation purposes as previously described (*Zhang et al., 2015*). Following surgical isolation of ganglia, they were placed in ice cold collection media composed of 124.5 mM NaCl, 5 mM KCl, 1.2 mM MgSO4, 1 mM CaCl2, and 30 mM HEPES, and had been filter sterilized after the pH had been adjusted to 7.35 with NaOH. As with our previous study, the time between cross-clamp and the harvest of ganglia was generally under 45 min, and the time between tissue collection and initiating the dissociation protocol was less than three hours. The same protocol and combination of solutions was employed in the present experiments, as described previously, except that the complete media used for plating the neurons consisted of basal media (500 ml bottle of L-15 media containing: 60 mg imidizole, 15 mg aspartic acid, 15 mg glutamic acid, 15 mg cystine, 5 mg $\beta$-alanine, 10 mg myo-inositol, 10 mg cholineCl, 5 mg p-aminobenzoic acid, 25 mg fumaric acid, 2 mg vitamin B12 and 5 mg of lipoic acid (which was first dissolved in methanol at a concentration of 1g/2.5 ml)) diluted with fetal bovine serum (1:10) and then supplemented to yield a final concentration of 50 ng/ml nerve growth factor (NGF 2.5S, Invitrogen), 0.3 mg/ml glutamine (Invitrogen), 4.5 mg/ml glucose (Sigma-Aldrich), 0.525 mg/ml ascorbic acid (Sigma-Aldrich), 2.4 μg/ml glutathione (Invitrogen), and 0.2% (w/v) NaHCO3 (Sigma-Aldrich). Cells were plated onto poly-L-lysine coated glass coverslips (Invitrogen) placed in 35 mm culture dishes and stored in a CO2 (5%) incubator at 37°C for 2–4 hr prior to flooding the culture dishes with Complete Media. Neurons were studied within 12 hr (acute) of plating.

## Isolation and plating of rat sensory neurons

Adult rat sensory neurons were surgically obtained, enzymatically treated and mechanically dissociated as previously described (*Lu et al., 2006*). Cells were also plated on poly-L-lysine coated glass coverslips (Invitrogen), which were placed in 35 mm culture dishes and stored in a C02 (3%) incubator at 37°C for 2 hr prior to flooding with Complete Media. Neurons were studied within 12 hr of plating.

## Whole cell patch clamp

Whole-cell patch-clamp recordings were performed with an Axopatch 200B controlled with pClamp (v 10.2) software (Molecular Devices, Carlsbad, CA) used in combination with a Digidata 1320A A/D converter (Molecular Devices). Unless otherwise noted, data were acquired at 20 kHz and filtered at 5 kHz. Borosilicate glass (WPI, Sarasota, FL) electrodes were 0.75–2 MΩ when filled with an electrode solution that contained (in mM): Cs- Methansulphonate 100, TEA-Cl 40, NaCl 5, CaCl$_2$ 1, EGTA 11, HEPES 10, Mg-ATP 2, and GTP 1; pH was adjusted to 7.2 with Tris-base and osmolality was adjusted to 310 mOsm with sucrose. The bath solution consisted of (in mM): NaCl 35, Choline-Cl 65, TEA-Cl 30, CaCl2 0.1, MgCl2 5, CdCl2 0.1, HEPES 10, and glucose 10; pH was adjusted to 7.4 with Tris-Base, and the osmolality adjusted to 320 with sucrose.

Capacitive currents were minimized with amplifier circuitry. Series resistance compensation was always employed, and if it was not possible to achieve compensation greater than 75%, neurons were not included for further analysis. Similarly, if estimated voltage errors were greater than 5 mV, data were not included for further analysis, where voltage errors were estimated based on the peak inward current across the uncompensated series resistance. Data was also excluded from neurons in which the holding current was >500 pA. Preliminary experiments indicated that it was rarely possible to maintain clamp control of currents evoked in neurons in culture for more than 24 hr, even with extracellular Na$^+$ reduced to 20 mM. Thus, all data included in this data set were from neurons <24 hr in culture. A p/−4 leak subtraction was employed from a holding potential of −80 mV. Steady-state availability curves were determined for each neuron in which both TTX-S and TTX-R currents were detected, so as to confirm the pre-pulse potential amplitude necessary for relief of steady-state inactivation, as well as the pre-pulse potential at which TTX-S currents were completely inactivated

(*Figure 1*). A 500 ms pre-pulse was used to drive changes in channel availability, followed by a voltage step to a potential that enabled visualization of both TTX-S and TTX-R components of the total current (generally between −5 and 0 mV). The pre-pulse was increased by 10 mV increments every 5 s. The pre-pulse potentials needed for full channel availability and for inactivation of TTX-S currents were used for the generation of current-voltage (I-V) curves for total current, and for TTX-R current. I-V curves were generated with a series of 15 ms test pulses between −60 and +40 mV, evoked every 5 s. It was then possible to subtract TTX-R currents from total current to obtain TTX-S currents in isolation (*Figure 1*). We subsequently confirmed that currents isolated in this manner were identical to those isolated with TTX (*Figure 1*).

## Reagents

Unless otherwise noted, all reagents were obtained from Sigma-Aldrich. TTX was dissolved in distilled water (dH2O) as a 1 mM stock solution, and stored at 4°C until use. A-803467 was dissolved in dimethylsulfoxide (DMSO) at a concentration of 10 mM, immediately before use, and subsequently diluted in bath solution. Bradykinin was dissolved in 0.1% acetic acid as a 10 mM stock solution, and stored at −20°C until use. Prostaglandin E2 was dissolved in ethanol as a 10 mM stock solution and stored at −20°C until use. Histamine was dissolved in dH2O as a 10 mM stock solution and stored at −20°C until use. Lidocaine was prepared as a 10 mM stock solution with 5 mM Na-HEPES buffered dH2O, with a pH adjusted to 7.0 with TEA-OH, and stored at −20°C until use.

## Data analysis

Amplifier circuitry was used to estimate membrane capacitance which was used to estimate current density. $Na^+$ current reversal potential ($V_r$) was determined for each neuron from the linear phase of the I-V curve. Conductance-voltage curves (G-V) were determined by dividing current evoked at each test potential ($V_t$) by the driving force on the current ($V_r – V_t$). A two-pulse protocol was used to determine the recovery from inactivation, where the first pulse to 0 mV, was used to completely inactivate $Na^+$ currents, and the second pulse to −50 mV or −90 mV of increasing duration was used to drive recovery of TTX-R and TTX-S currents from inactivation, respectively. The second pulse was followed by a final test pulse to 0 mV. Steady-state inactivation and G-V curves were fitted with modified Boltzmann equations so as to determine maximal conductance (Gmax), the voltage at which current were either half inactivated or half activated ($V_{0.5}$), as well as the slopes of the two curves. Recovery from inactivation data were fitted with a double exponential. Current activation, inactivation, and when possible, deactivation, were determined with a single exponential fitted to the rising phase (activation), falling phase (inactivation), and tail currents (deactivation) of currents evoked with an I-V protocol. Use-dependent block was determined with 20 pulses to 0 mV delivered at 1, 2, or 5 Hz. Fractional use-dependent block was estimated by dividing current evoked after the 20th pulse by the current evoked after the first. Finally, concentration-response curves were fitted with a modified Hill equation to enable estimation of the concentration needed to block 50% of evoked current.

A t-test was used for statistical comparisons between human and rat data, where a difference with $p < 0.05$ was considered statistically significant. If comparisons were made between human and rat across voltage, a two-way ANOVA was used. Prior to use of a t-test of ANOVA, data to be compared was assessed for normality and equal variance. Linear regression and a one-way ANOVA were used to assess the potential impact of age on evoked currents.

## Acknowledgements

This work was supported in part by a Grant from Eli Lilly, as well as Grants from the National Institutes of Health (R01DE018252).

## Additional information

### Competing interests

BTP: Employee of Eli Lilly. MSG: Has received grant support from Eli Lilly and Grunenthal and has served on an advisory panel for Grunenthal and the Global Pain Foundation. The other authors declare that no competing interests exist.

## Funding

| Funder | Grant reference number | Author |
|---|---|---|
| National Institutes of Health | R01DE018252 | Michael S Gold |
| Eli Lilly and Company | | Michael S Gold |

The funders had no role in study design, data collection and interpretation, or the decision to submit the work for publication.

## Author contributions

XZ, Data curation, Formal analysis, Methodology, Writing—original draft, Writing—review and editing; BTP, Conceptualization, Methodology, Writing—review and editing; IB, Conceptualization, Project administration, Writing—review and editing; MSG, Conceptualization, Resources, Data curation, Software, Formal analysis, Supervision, Funding acquisition, Validation, Investigation, Methodology, Project administration, Writing—review and editing

## Author ORCIDs

Michael S Gold, http://orcid.org/0000-0002-2083-6206

## Ethics

Human subjects: DRG were collected from organ donors with the consent of family members for the use of their loved one's tissue for research purposes. The protocol for the collection and study of tissue from organ donors was approved by the University of Pittsburgh Committee for Oversight of Research and Clinical Training Involving Decedents. CORID ID #358

Animal experimentation: All procedures were approved by the University of Pittsburgh Institutional Animal Care and Use Committee and performed in accordance with National Institutes of Health guidelines for the use of laboratory animals in research. IACUC Protocol #12121265

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
