## [Decision Letter]

Thank you for submitting your article "Voltage-Gated Na^+^ Currents in Human Dorsal Root Ganglion Neurons" for consideration by *eLife*. Your article has been reviewed by two peer reviewers, and the evaluation has been overseen by a Reviewing Editor and Gary Westbrook as the Senior Editor. The following individuals involved in review of your submission have agreed to reveal their identity: Bruce P Bean (Reviewer #1); Stephen Cannon (Reviewer #2).

The reviewers have discussed the reviews with one another and the Reviewing Editor has drafted this decision to help you prepare a revised submission.

Summary:

This manuscript describes the biophysical properties and pharmacology of voltage-dependent sodium channels in dorsal root ganglion neurons from adult humans and provides a comparison of the characteristics of those sodium currents to the more common experimental model of rats. The work identifies both similarities and differences between the human and rat preparations.

Essential revisions:

Both reviewers found the work to be very important, informative, and well done. Their compiled favorable comments are quoted verbatim below in "General Comments." Several key points came up that require addressing, however, relating to pharmacology, which was seen as the weakest area of the manuscript, and a few other aspects of voltage protocols and cell selection. The points that require addressing are summarized here and elaborated upon in the reviewers' words under "Detailed Points for Essential Revisions."

1) Verify that the very surprising results indicating that the primary TTX-S channels in humans (a) are not 1.7 and (b) are not blocked by the broad-spectrum Protoxin II are not influenced by plausible sources of error, e.g., those relating to complications of peptide perfusion.

2) Omit or address the very unexpected result that TTX increases ramp current, which also raises questions about the efficacy/specificity of drug perfusion.

3) Clarify voltage protocols used and (a) use the information to strengthen/clarify/curtail conclusions based on pharmacology and (b) validate (test and/or rationalize) the prepulse approach as opposed to a pharmacological approach to separate the currents in humans.

4) Test and/or clarify ambiguities arising from size differences and size-based selection to verify that differences between species weren't based on clamp artifacts from the different sizes of neurons.

5) Address or omit the deactivation time constant measurements, about which specific questions were raised.

We recognize that the availability of human tissue may be limited. If it is not possible or practical to address the issues directly, a (less desirable) alternative is to tone down or qualify the conclusion that Nav1.7 may not contribute much to the TTX-S current, indicating that further pharmacological work is necessary for a definitive conclusion.

General comments

This is important experimental information, because almost all previous knowledge about the properties and functional roles of sodium channels in DRG neurons has come either from studies on rodent DRG neurons or on cloned human channels in heterologous expression systems. Since an important goal of the research area, especially on the pharmacology, is to help guide development of new treatments for pain, possible differences in the pharmacology of native human channels compared to those in rodent neurons or to human channels in heterologous systems are of great interest. Consequently, the surprising results in this manuscript suggesting that the pharmacological properties of both TTX-resistant and TTX-sensitive populations of sodium channels in human neurons differ from those in rat neurons give the manuscript considerable interest. Beyond this, the work shows, very surprisingly, that obtaining electrophysiologically healthy neurons acutely-dissociated from human DRGs is actually easier (or at least has a higher success rate once the DRGs are obtained) than with rat DRG neurons of fully adult animals. The technical quality of the voltage-clamp studies in the manuscript are excellent and the volume of data for many of the figures leaves no doubt of the possibility of obtaining statistically impressive data from human neurons. Thus an important secondary message of the manuscript will be to encourage others to study human DRG neurons.

An excellent feature of the work is a detailed comparison with the properties of rat DRG neurons studies with the same protocols. This gives confidence that the differences seen are genuine and not the result of slightly different protocols (although as noted below, it would helpful to have a statement about the extent to which the experiments were performed in parallel, especially in regard to the pharmacologic experiments).

In general, the biophysical description of both the TTX-R and TTX-S components of sodium current are thorough and convincing (although as noted below, it would be helpful to have a more complete description of some of the voltage protocols in some cases).

The repertoire of voltage gated sodium channels (VGSCs) in dorsal root ganglia (DRG) and their modulation by inflammatory mediators are important for the perception of pain and the hypersensitivity of pain syndromes associated with tissue injury. Much of this knowledge has come from studies in rodent DRG neurons or from human VGSCs expressed in heterologous systems. This practical constraint leaves open the question of whether the VGSC properties of human DRG neurons are comparable. Zhang and colleagues have recorded Na currents from human DRG neurons acutely isolated from organ donors. In broad terms, the two major classes of Na currents (TTX-R and TTX-S) were both present and comparable in rat and human DRG neurons. Several differences, however, were identified and these may have important consequences for the pathogenesis of pain syndromes and for therapeutic strategies. For example, the putative NaV1.8-selective blocker, A803467, did not cause a detectable reduction of human TTX-R currents until 1 μm was applied, whereas rat TTX-R currents are much more sensitive to block. Human TTX-R currents also had no evidence of use-dependent cumulative inactivation or use-dependent lidocaine block (1-5 HZ) whereas rat TTX-R current had substantial use dependence for both. Differences of intrinsic gating were also observed. Most striking were faster activation and recovery from inactivation for human TTX-S currents as compared to rat TTX-S. Overall, this is a carefully designed and conducted study that demonstrates while the VGSC properties in DRG neurons are overall similar for rat and human, there are clear differences that may have important implications for development of therapeutics.

Detailed points for essential revisions:

1) The area in which the results in the manuscript could be stronger is the most important – the conclusion that the TTX-S channels in the human neurons are not primarily carried by Nav1.7 channels. If true, this is very important, since multiple companies are developing Nav1.7 inhibitors with the assumption that these channels are critical for the function of nociceptors. Also, the conclusion that Nav1.7 do not contribute to the TTX-S currents is opposite to a previous study of human DRG neurons in which it was concluded that the majority of the TTX-S current was carried by Nav1.7 channels, based on inhibition of 75% of the current by 30 nM PF-05089771, a fairly selective blocker of Nav1.7 channels. The authors point out this previous study used neurons in culture for up to 9 days, which could cause an upregulation of Nav1.7 channels – a valid point. They also suggest that the result could have been from imperfect selectivity of PF-05089771, which while possible seems a lot less likely, since Alexandrou et al., 2016 saw big effects of 30 nM PF-05089771, which would be expected to have minimal effects on any of the other channel types.

Here the authors present two pieces of evidence arguing against mediation of TTX-S by Nav1.7. One is the lack of effect of 10 nM Protoxin II. It seemed very puzzling that the authors chose protoxin II as a blocker, because a previous review of the properties of this toxin stated "ProTx-I and ProTx-II inhibit all sodium channel (Nav1) subtypes tested with similar potency." (Priest et al. Toxicon. 2007 49:194-201). This is certainly not clear from the presentation, where a reader would naturally infer that the toxin was used because it had selectivity for Nav1.7. The evidence against Nav1.7 would be far stronger if the authors had used PF-05089771, which is much more selective. This compound has been commercially available (from both Σ and Tocris) for some time.

Given the efficacy of protoxin-II against all sodium channels, the lack of effect is very surprising whatever channels underlie the TTX-S current. A possible experimental problem with potent peptide toxins of this type is that they can bind to plastic tubing and glassware and be difficult to remove, so that subsequent application appears to have no effect because of presence of the agent under "control" conditions. I raise that because the lack of any effect of the toxin is unexpected whatever channels the sodium currents are coming from and because we have experienced that insidious problem several times in my own lab with similar toxins – a problem that could only be resolved by completely replacing tubing and chambers.

2) The second piece of evidence that authors present against the contribution pf Nav1.7 channels is the most puzzling result in the paper – that TTX not only did not inhibit a current induced by ramp of voltage but actually enhanced the current and shifted its voltage dependence to more hyperpolarizing voltages. The shift of current to more hyperpolarized voltage makes no sense at all and suggests some sort of experimental problem. It would suggest that TTX enhances sodium current for small depolarizations which has never been described for any channels. At a minimum, the authors need to follow up the ramp experiment with step depolarization experiments to characterize this effect in more detail. Even if TTX simply had no effect on the ramp current, this is only weak evidence against mediation by Nav1.7. It could reflect for example faster or more complete closed-state inactivation in human Nav1.7 channels versus rodent Nav1.7 channels, which would be interesting. The authors did not compare closed-state inactivation in the human vs rat neurons so this seems like an open possibility.

3) It would be helpful if all the figures had clear statements of exactly what the voltage protocols were, which were not always clear to me. In Figure 4–Figure 7 on TTX-R current, it seems that the records were obtained from a steady holding voltage of -80 mV and a 500 ms prepulse, probably to -40 mV based on Figure 1, but this is not stated explicitly and the prepulse potential is not given in each case. Materials and methods says "The pre-pulse potentials needed for full channel availability and for inactivation of TTX-S currents were used for the generation of current-voltage (I-V) curves for total current, and for TTX-R current" which suggests that the same prepulse was not always used. The exact protocol in each figure should be stated. This is especially an issue for interpreting the effects of lidocaine, since the potency is so strongly sensitive to resting membrane potential. It seems possible that the potency of A-803467 could also be sensitive to the exact voltage protocol.

Additionally, the separation of TTX-R and TTX-S components by using a conditioning voltage pulse (instead of the more laborious method of TTX block) is a key aspect of the analysis and interpretation. Deficiencies in this separation technique, which may be different for rat versus human DRG neurons, could result in apparent differences or underestimate differences. With that in mind, how was it possible to determine the availability curve for TTX-R specific currents in Figure 4? The conditioning pulse range extended from -60 mV to 0 mV and yet Figure 1 shows a prepulse of -35 mV is needed to isolate the TTX-R component (via inactivation of TTX-S). How was availability data obtained for conditioning voltages more negative than -35 mV?

A great feature of the work is the comparison to studies in rat neurons done in the same lab. If these were actually done in parallel, using for example exactly the same batches of pharmacological reagents and by the same experimenter, this might be mentioned explicitly as it further strengthens the point that the differences are genuine and not the result of differences in procedures or reagents.

4) A missing piece of information is exactly how the authors selected which cells to record from. From the fact that they all had TTX-R current, the experimenter presumably chose smaller-diameter cells to record from, but what exactly was the selection criterion? Do the authors think the cells all correspond to cell bodies of C-fibers? The authors have discussed the challenges of high Na current density and series resistance that often plague studies of VGSCs. Because human DRG soma are much larger than rat, the demands on clamp quality would be systematically more challenging in recordings from human than rat neurons. This might cause a systematic difference in the apparent gating behavior. The distribution of cell capacitance did, however, have a region of overlap around 50 pF. Perhaps a subset of rat and human DRG neurons with comparable size could be analyzed to provide reassurance that the observed species differences were still detectable in a population of similar-sized neurons?

5) The interpretation and discussion of VGSC deactivation was confusing (Figure 5, which shows plots of deactivation tau versus voltage). First, deactivation time constants are usually presented as a function of the deactivation pulse potential, not the preceding voltage to activate the channel as shown in Figure 5. However, from the text it seems that this is not a standard plot of tail time constant against voltage as one might assume at first but rather a measurement of the tail time constant at -60 mV following steps to different voltages. This should be made clear in the figure legend. Were there any measurements of deactivation time constants at other voltages? These would be helpful to someone who wanted to make a Hodgkin-Huxley type model from the data. Second, deactivation rates are usually measured over a range of test potentials. The more hyperpolarized ones are often regarded as more representative of "true" deactivation (O->C transition) but the fast kinetics are technically demanding. Responses at more depolarized potentials are more easily resolved, but may represent other state transitions (not just O -> C). The canonical view is that a true deactivation rate should be independent of the pulse potential used to open the channel. The atypical behavior in Figure 5 suggests a pulse to -60 mV was not measuring deactivation in isolation. Moreover, deactivation data are not presented for TTX-S currents. Perhaps the deactivation data should be omitted?

[Editors' note: further revisions were requested prior to acceptance, as described below.]

Thank you for resubmitting your work entitled "Voltage-Gated Na^+^ Currents in Human Dorsal Root Ganglion Neurons" for further consideration at *eLife*. Your revised article has been evaluated by Gary Westbrook (Senior editor), a Reviewing editor, and two reviewers.

The manuscript has been improved but there are some remaining issues that need to be addressed before acceptance, as outlined below:

"TTX-S" and "TTX-R" are used as shorthand for prepulse subtraction, which is validated for control solutions in Figure 1. This procedure may complicate the interpretation of some of the results, however, particularly the pharmacology leading to the conclusion that 1.7 is not the primary TTX-S channel. Please edit the text (especially the Introduction and Discussion) for the following:

1) To make sure that readers are clear that "TTX-S" and "TTX-R" actually refer to currents inactivated at -40 mV and available at -40 mV,

2) To ensure that the prepulse subtraction procedure is kept in mind in interpreting the effects of the blockers, and

3) To limit or constrain claims that NaV1.7 is not the primary TTX-S current, as appropriate.

Please also include a statement about informed consent (as appropriate) for collection of human tissue.

The full comments pertaining to the requested revisions are below and have been agreed upon by all participants in the review.

General assessment and major comments:

The most important change in the manuscript is new data with the Pfizer Nav1.7 inhibitor PF-05089771, which the authors find inhibits the TTX-S current quite potently, with an IC50 of about 10 nM. In addition, the authors have clarified some the details of the voltage protocols and cell selection.

I was very surprised that with the new data showing potent block of the TTX-S current by PF-05089771, the authors still end up concluding that Nav1.7 channels make little contribution to the TTX-S current in human DRG neurons. If it were my data, my interpretation of the overall set of the results the authors present would be the opposite, that the TTX-S current is likely to be primarily from Nav1.7 channels based on its sensitivity to PF-05089771, but that it is (interestingly) less sensitive to Protoxin-II than rat Nav1.7 channels. The observation that PF-05089771 also seems to inhibit TTX-R currents in human DRG neurons at higher concentrations is interesting and important. So, like most small molecules, the selectivity of PF-05089771 is imperfect. That is important to know, and constitutes another of the many interesting points in the manuscript about different pharmacology of the native currents in human DRG neurons compared to rat DRG neurons or heterologously-expressed human channels. However, it is still true that PF-05089771 inhibits the TTX-S current with significantly higher potency, and about what is expected from the effects on cloned human Nav1.7 channels. So my interpretation would be that the results with PF-05089771 imply that the TTX-S current is mostly from Nav1.7, but that the compound is less selective than previously realized and at higher concentrations also inhibits native human Nav1.8 channels – a very important point that adds to the significance of the results in the manuscript.

To me, the flow of the revised manuscript now seems strange in that final set of experimental results in Results showing potent block by PF-05089771 gives the reader the impression that the TTX-S current is likely mainly Nav1.7, but then in the Discussion the first mention of this issue suggests the opposite without explaining why. The first part of the Discussion summarizing the pharmacology is very reasonable and straightforward (the paragraph starting, "With respect to drug development, the results of the present study are consistent with those from previous studies indicating that both expression systems and species differences may have a significant influence on the pharmacological properties of the protein in question." This paragraph nicely summarizes why it is not shocking that the pharmacology of channels might differ between rat and human cells and between native currents and those from heterologous expression.

The first suggestion in the Discussion that the authors conclude that Nav1.7 does not make the major contribution to TTX-S current comes obliquely, in a sentence in a paragraph about the up-regulation of currents by the inflammatory soup "As noted above, data from rodent DRG neurons suggest that NaV1.7 accounts for ~70% of the TTX-S current in small diameter neurons and NaV1.6 accounts for the remainder (Laedermann et al., 2013). However, the suggestion that the contribution of NaV1.7 varies across subpopulations of human sensory neurons is also consistent with the suggestion that NaV1.7 is not the dominant, let alone only VGSC subtype in human sensory neurons." In the original version, this sentence referred to a stated conclusion in Results based on the protoxin-II results that Nav1.7 was not the dominant current. But in the revised version, there has been no explanation yet of why the authors would conclude this, and it seems to come out of nowhere.

The authors' argument about Nav1.7 comes later, in the paragraph starting "The possibility that NaV1.7 may not be the dominant subunit underlying TTX-S currents, let alone significantly contribute to these currents in human DRG neurons, as suggested by our inflammatory mediator results, was further supported by our results with Pro-Tx II and PF-05089771." This is where the authors argue that the block by PF-05089771 is not good evidence for Nav1.7 mediating the TTX-S because they find that PF-05089771 at higher concentrations also inhibits the TTX-R current. To me, the argument in this paragraph was not convincing, as it essentially argues that the lack of effect of protoxin-II is more convincing than the potent block by PF-05089771. (The lack of ramp currents from TTX-S currents is another argument, but as the authors point out when presenting this data, this could be easily explained by different gating kinetics between human and rat TTX-S currents.).

There is a reason to be cautious about interpreting the results with PF-05089771 and Protoxin-II, because both interact with their target channels in a state-dependent manner. The authors did not study the effects of the blockers on isolated currents from TTX-R or TTX-S channels, but rather used a prepulse procedure to distinguish them by inactivation. They show that this protocol works in the absence of drugs, but obviously it may not in the presence of drugs that can modify the voltage-dependence of inactivation. In my view, that should introduce a lot of caution in interpreting these results. In fact, in reading the paper quickly, it would be very easy for a reader to miss the important point that when the authors refer to "TTX-S" and "TTX-R" currents, this is really shorthand for "currents available from -80 but inactivated by a prepulse to -40 mv" and "currents remaining after a prepulse to -40 mV". Although the authors clearly state this in the course of Results, I think it is an important point to reiterate in the Discussion, because it is an important qualification for interpreting the pharmacological results (not only for protoxin and PF-05089771, but also lidocaine and A-803467, which are also state-dependent blockers).

Because of these issues, I did not find the authors' suggestion that Nav1.7 accounts for little if any of the TTX-S current convincing and it would not be my interpretation. Of course, the authors should be free to interpret their own data, but they should probably consider at least qualifying their conclusion.

Whatever the interpretation given, the experimental results are very interesting and leave no doubt that the detailed pharmacology of the channels in human cells is quantitatively different in interesting ways compared to the currents in rat neurons, probably most importantly in the relative lack of use-dependence with lidocaine and the apparent significant sensitivity of TTX-R channels to PF-05089771.

I did not see a statement about informed consent for the collection of the human tissue, which seems important. Presumably this is the authors' 2015 paper that is cited for the Methods, but it seems that it should be here also.

---

## [Author Response]

*Essential revisions:*

*Both reviewers found the work to be very important, informative, and well done. Their compiled favorable comments are quoted verbatim below in "General Comments." Several key points came up that require addressing, however, relating to pharmacology, which was seen as the weakest area of the manuscript, and a few other aspects of voltage protocols and cell selection. The points that require addressing are summarized here and elaborated upon in the reviewers' words under "Detailed Points for Essential Revisions."*

*1) Verify that the very surprising results indicating that the primary TTX-S channels in humans (a) are not 1.7 and (b) are not blocked by the broad-spectrum Protoxin II are not influenced by plausible sources of error, e.g., those relating to complications of peptide perfusion.*

We agree with reviewers that this is an important concern, and as noted below, have addressed this issue with additional data and further discussion. The additional data include results from the application of Protoxin II to rat DRG neurons, as well as data from human DRG neurons in response to the putative NaV1.7 selective small molecule channel blocker PF-05089771. With respect to the former, we were able to demonstrate that Protoxin II blocked a fraction of the TTX-S current in rat DRG neurons, arguing against potential complications with handling the peptide. With respect to the latter, we were able to demonstrate complete block of TTX-S currents in human DRG neurons with PF-05089771 with an IC50 of ~6 nM, remarkably close to that reported by Alexandrou and colleagues (Alexandrou et al., 2016). However, in marked contrast to heterologous expression results reported by Alexandrou and colleagues, we observed a significant reduction in TTX-R current in human DRG neurons treated with PF-05089771, with an IC50 of ~50 nM. Given that Alexandrou and colleagues reported that heterologously expressed NaV1.8 was resistant to PF-05089771 at concentrations as high as 10 μM, these observations underscore the need to validate results from heterologous expression studies on native proteins expressed in their native environment. The observation that PF- 05089771 has limited utility for the differential block of TTX-S and TTX-R currents in human DRG neurons raises the possibility that this compound also has limited utility for the differential block of NaV1.7 relative to other TTX-S channels likely present in human DRG neurons. Thus, we would argue that when considered in the context of our results with Protoxin II and our ramp current data, the limited selectivity of PF-05089771 on native channels lends further support to the suggestion that NaV1.7 is not the only, or even dominant channel in human DRG neurons.

We have expanded our discussion of these issues to address several points. The first concerns the selectivity of Protoxin II. As noted below, the reviewers are correct to note the sentence in the 2007 Priest paper, in which the authors stated that the toxin blocked all NaV1 isoforms (Priest et al., 2007). However, the Priest et al., 2007 paper represents a literature review of limited available data. Importantly, they went on to characterize this toxin in more detail, reporting in 2008, that the toxin was at least 100 time more potent against NaV1.7 than other NaV1 isoforms (Schmalhofer et al., 2008). Thus, in our revised manuscript, we acknowledge that our negative results with this toxin in human DRG neurons may have been due to technical problems, and the results with PF- 05089771 may still reflect a significant contribution of NaV1.7 to the TTX-S current in human DRG neurons. However, we also suggest an alternative hypothesis, which is that NaV1.7 is not the primary channel underlying TTX-S currents in human DRG neurons, where the negative results with Protoxin II and the altered pharmacological profile of PF- 05089771 reflect the impact of the native environment on the pharmacological properties of the channels present. Consistent with the suggestion that the relative contribution of NaV1.7 to TTX-S currents in rodent sensory neurons may have been over-estimated are the recent results from Deuis and colleagues (2017), who reported that a putative NaV1.7 selective spider venom peptide blocked no more than 50% of the TTX-S current in every subpopulation of sensory neurons tested (Deuis et al., 2017). A relatively limited role for NaV1.7 in human DRG neurons would also be consistent with our observation that a low threshold ramp current is absent in human DRG neurons as well as the relatively limited impact gain of function mutations in NaV1.7 have on the excitability of human nociceptive afferents (Namer et al., 2015).

*2) Omit or address the very unexpected result that TTX increases ramp current, which also raises questions about the efficacy/specificity of drug perfusion.*

We agree with the reviewers that this was a very perplexing observation. Per the reviewer’s suggestion, we have omitted these data. We still describe the high threshold ramp current in these neurons, however.

That said, we have pursued this issue further in the literature, and came across a couple of relevant studies. Most relevant is a study by Farmer and colleagues (2008), who observed a TTX-induced increase in TTX-R currents in rat DRG neurons that was associated with a leftward shift in the voltage-dependence of activation (Farmer et al., 2008). These authors concluded that the modulation of TTX-R currents by TTX was due to the relief of a tonic block of these channels by La3+ (used to block Ca^2+^ currents in their recording solution). We used Cd2+ rather than La3+ to block Ca^2+^ currents in our experiments, but there is also evidence of a Cd2+-induced block of TTX-R currents in DRG neurons (Kuo et al., 2002). Thus, it is possible that the leftward shift in the ramp current we observed was due to a TTX-induced relief of Cd2+ block. Importantly, such an explanation would support the conclusion that ramp currents in human DRG neurons are largely carried by TTX-R currents. If the reviewer’s agree that the evidence is sufficient to support this speculation, we would happily include our TTX ramp results in the manuscript, but leave this decision to the reviewers/editor.

*3) Clarify voltage protocols used and (a) use the information to strengthen/clarify/curtail conclusions based on pharmacology and (b) validate (test and/or rationalize) the prepulse approach as opposed to a pharmacological approach to separate the currents in humans.*

We have provided additional details about the voltage protocols used throughout the study. We have acknowledged the possibility that the use of a voltage- protocol to isolate TTX-S and TTX-R currents introduced a source of error in the characterization of current properties, particularly for pharmacological experiments where the drugs used could have influenced the biophysical properties of the channels such that isolation of current with voltage-steps was no longer possible. We have also added further justification for our decision to rely on voltage protocols for current separation. This decision was primarily based on the facts that 1) there was a finite time- frame over which stable recordings were obtained, 2) the use of a voltage-protocol was faster than a pharmacological approach, at least with respect to the reversal of the pharmacological block of TTX-S channels, and 3) we wanted to maximize the amount of data collected from each neuron. With respect to this last point, the use of voltage protocols to isolate TTX-S and TTX-R currents enabled us to monitor both currents in each neuron.

*4) Test and/or clarify ambiguities arising from size differences and size-based selection to verify that differences between species weren't based on clamp artifacts from the different sizes of neurons.*

This is an important point. We have added a section to the revised manuscript specifically addressing this concern. We have laid out the data arguing against clamp artifacts on the biophysical parameters obtained. We have also acknowledged the limitations associated with assumptions about the phenotype of the human neurons.

*5) Address or omit the deactivation time constant measurements, about which specific questions were raised.*

Done

*We recognize that the availability of human tissue may be limited. If it is not possible or practical to address the issues directly, a (less desirable) alternative is to tone down or qualify the conclusion that Nav1.7 may not contribute much to the TTX-S current, indicating that further pharmacological work is necessary for a definitive conclusion.*

The reviewer’s suggestion about the use of other NaV1.7 blockers prompted us to go back into the analysis of compounds that we originally tested for Eli Lilly when we got this project up and running. We were blinded to the compounds used, but in fact had a complete data set with PF-05089771, which included the activity of this compound on TTX-R currents in human DRG neurons, an observation not previously reported.

Importantly, and consistent with our original suggestion about the interpretation of the previous data with this compound on human DRG neurons, this drug does not appear to be as selective on native channels in human neurons as it was reported to be against heterologously expressed channels. We have also included additional data with protoxin II against rat TTX-S currents. Thus, while we have now acknowledged the limitations of the conclusions drawn from our pharmacological and biophysical data, we suggest that our results in the context of data available in the literature strongly support the possibility that NaV1.7 is not the dominant channel underlying TTX-S currents in human DRG neurons.

[…]

*Detailed points for essential revisions:*

*1) The area in which the results in the manuscript could be stronger is the most important – the conclusion that the TTX-S channels in the human neurons are not primarily carried by Nav1.7 channels. If true, this is very important, since multiple companies are developing Nav1.7 inhibitors with the assumption that these channels are critical for the function of nociceptors. Also, the conclusion that Nav1.7 do not contribute to the TTX-S currents is opposite to a previous study of human DRG neurons in which it was concluded that the majority of the TTX-S current was carried by Nav1.7 channels, based on inhibition of 75% of the current by 30 nM PF-05089771, a fairly selective blocker of Nav1.7 channels. The authors point out this previous study used neurons in culture for up to 9 days, which could cause an upregulation of Nav1.7 channels – a valid point. They also suggest that the result could have been from imperfect selectivity of PF-05089771, which while possible seems a lot less likely, since Alexandrou et al., 2016 saw big effects of 30 nM PF-05089771, which would be expected to have minimal effects on any of the other channel types.*

*Here the authors present two pieces of evidence arguing against mediation of TTX-S by Nav1.7. One is the lack of effect of 10 nM Protoxin II. It seemed very puzzling that the authors chose protoxin II as a blocker, because a previous review of the properties of this toxin stated "ProTx-I and ProTx-II inhibit all sodium channel (Nav1) subtypes tested with similar potency." (Priest et al. Toxicon. 2007 49:194-201). This is certainly not clear from the presentation, where a reader would naturally infer that the toxin was used because it had selectivity for Nav1.7. The evidence against Nav1.7 would be far stronger if the authors had used PF-05089771, which is much more selective. This compound has been commercially available (from both Σ and Tocris) for some time.*

The reviewer raises important points. We had, in fact tested PF-05089771 on human DRG neurons and observed a complete block of TTX-S currents with 30 minutes of incubation with an IC50 of 6 nM. However, the IC50 for the block of TTX-R currents was 50 nM. Admittedly, these data support the possibility that NaV1.7 does in fact underlie the majority of TTX-S currents in human DRG neurons. However, given that heterologous expression data in the Alexandrou paper indicated that heterologously expressed NaV1.8 is insensitive to PF-05089771 at concentrations as high as 10 μM, our data with this compound at support the possibility that the TTX-S current blocked reflects more than activity in NaV1.7. Of note, Alexandrou and colleagues studied TTX-S currents in the presence of A-803467 to block NaV1.8, and therefore would not have detected an influence of PF-0589771 on TTX-R currents. These data have been added to the revised manuscript.

By way of an explanation for our failure to include results with PF-05089771 in our original manuscript, these data were collected in a series of experiments in which other proprietary compounds from Eli Lilly were also tested. We had not broken the code on the compounds assessed until this question was raised (results were submitted to Lilly in the blinded manner in which they were obtained). These data have now been added to the revised manuscript.

On the other hand, the choice of Pro-Tx II was based on evidence that this toxin is a potent and selective blocker of NaV1.7 as documented by Schmalhofer and colleagues (Schmalhofer et al., 2008), and subsequently others. The toxin is sold as an NaV1.7 selective blocker through companies such as Σ and Tocris. The Priest et al. 2007 paper referred to contains a sentence in the Abstract that is admittedly misleading, based largely on the results of a study by Middleton and colleagues (2002), in which results from variety of assay protocols were combined (Middleton et al., 2002). The Schmalhofer paper was a follow-up to this 2007 review which contained a more careful characterization of Pro-Tx II, in which the selectivity of the toxin was clearly documented (Schmalhofer et al., 2008).

The negative data with Pro-Tx II combined with the considerably lower level of selectivity of PF-05089771 in human DRG neurons than that reported in heterologous expression system and the absence of low threshold ramp currents in human DRG neurons all point to the possibility that NaV1.7 may not be the dominant TTX-S channel in human DRG neurons. This suggestion is further substantiated by the relatively minor impact of NaV1.7 gain of function mutations on the excitability of human nociceptive afferents documented in microneurography studies (Namer et al., 2015). While the activity of the spider venom peptide Pn3a on native TTX-R currents was not well described, recent results with this putatively NaV1.7 selective blocker would also suggest that NaV1.7 does not even underlie the majority of TTX-S current in rat DRG neurons (Deuis et al., 2017). It should also be noted that recent data from the NaV1.7 null mutant mice suggest that this subunit may, in fact play a more important role in mediating Na^+^ influx and associated changes in gene expression than it does in the electrical properties of these neurons (Minett et al., 2015), while a more prominent role in transmitter release from the central terminals (Alexandrou et al., 2016), may also contribute to a more limited role for this channel in the TTX-S currents detected in the afferent cell body.

Nevertheless, because these lines of evidence are indirect, these possibilities are only suggested in the revised manuscript.

*Given the efficacy of protoxin-II against all sodium channels, the lack of effect is very surprising whatever channels underlie the TTX-S current. A possible experimental problem with potent peptide toxins of this type is that they can bind to plastic tubing and glassware and be difficult to remove, so that subsequent application appears to have no effect because of presence of the agent under "control" conditions. I raise that because the lack of any effect of the toxin is unexpected whatever channels the sodium currents are coming from and because we have experienced that insidious problem several times in my own lab with similar toxins – a problem that could only be resolved by completely replacing tubing and chambers.*

We appreciate the reviewer’s suggestion, but tried a number of different strategies, including siliconizing glass tubes and the application of the toxin via siliconized glass pipettes. We suggest that the issue is really the concentration range over which the drug was tested. While we did use concentrations as high as 300 nM, the majority of experiments involved concentrations of 30 nM or lower. More importantly, we were able to demonstrate a block of TTX-S currents in rat DRG neurons of 50.5 ± 13.7 (n = 5)% with 10 nM toxin. Thus, while we have acknowledged that our failure to detect an effect of the toxin in human DRG neurons may reflect problems with working with the toxin, we do not think that was the primary source of the negative results obtained.

*2) The second piece of evidence that authors present against the contribution pf Nav1.7 channels is the most puzzling result in the paper – that TTX not only did not inhibit a current induced by ramp of voltage but actually enhanced the current and shifted its voltage dependence to more hyperpolarizing voltages. The shift of current to more hyperpolarized voltage makes no sense at all and suggests some sort of experimental problem. It would suggest that TTX enhances sodium current for small depolarizations which has never been described for any channels. At a minimum, the authors need to follow up the ramp experiment with step depolarization experiments to characterize this effect in more detail. Even if TTX simply had no effect on the ramp current, this is only weak evidence against mediation by Nav1.7. It could reflect for example faster or more complete closed-state inactivation in human Nav1.7 channels versus rodent Nav1.7 channels, which would be interesting. The authors did not compare closed-state inactivation in the human vs rat neurons so this seems like an open possibility.*

We completely agree with the reviewer that this was a perplexing and unexpected observation. There was no suggestion from the step depolarization data that TTX-R currents were facilitated by the presence of TTX. Nor was there evidence of a shift in the gating of TTX-R currents in the presence of TTX, as there was no significant change in either the V1/2 of activation (which changed 0.07 ± 0.04 mV after the application of TTX) or the slope of the GV curve (which changed by 0.03 ± 0.03 mV after the application of TTX). Furthermore, with a decrease in total current following application of TTX, a decrease in voltage error should have resulted in a depolarizing shift in the ramp current. We did not observe a similar shift in rat DRG neurons, suggesting that whatever the underlying mechanism, it was only detected in human neurons. Nevertheless, because we did not assess the onset of closed state inactivation, we have removed these data from the manuscript.

*3) It would be helpful if all the figures had clear statements of exactly what the voltage protocols were, which were not always clear to me. In Figure 4–Figure 7 on TTX-R current, it seems that the records were obtained from a steady holding voltage of -80 mV and a 500 ms prepulse, probably to -40 mV based on Figure 1, but this is not stated explicitly and the prepulse potential is not given in each case. Materials and methods says "The pre-pulse potentials needed for full channel availability and for inactivation of TTX-S currents were used for the generation of current-voltage (I-V) curves for total current, and for TTX-R current" which suggests that the same prepulse was not always used. The exact protocol in each figure should be stated. This is especially an issue for interpreting the effects of lidocaine, since the potency is so strongly sensitive to resting membrane potential. It seems possible that the potency of A-803467 could also be sensitive to the exact voltage protocol.*

We apologize for the confusion and have provided the information requested – at least for the raw traces included in each figure because the reviewer is correct, we did vary the protocol in some neurons to facilitate our ability to isolate TTX-S from TTX-R currents.

*Additionally, the separation of TTX-R and TTX-S components by using a conditioning voltage pulse (instead of the more laborious method of TTX block) is a key aspect of the analysis and interpretation. Deficiencies in this separation technique, which may be different for rat versus human DRG neurons, could result in apparent differences or underestimate differences. With that in mind, how was it possible to determine the availability curve for TTX-R specific currents in Figure 4? The conditioning pulse range extended from -60 mV to 0 mV and yet Figure 1 shows a prepulse of -35 mV is needed to isolate the TTX-R component (via inactivation of TTX-S). How was availability data obtained for conditioning voltages more negative than -35 mV?*

We apologize for failing to make this more clear. As can be seen in Figure 1, it was possible to use a combination of voltage and time to separate TTX-S from TTX-R currents. That is, in every neuron in which the TTX sensitivity of the TTX-S current was confirmed with TTX, the current evoked at potential ranging between -5 mV and +5 mV was completely inactivated by 10 ms after the start of the voltage step. Thus, it was possible to monitor TTX-R current availability across a full range of pre-pulse potentials. We have further clarified this point in the revised manuscript.

*A great feature of the work is the comparison to studies in rat neurons done in the same lab. If these were actually done in parallel, using for example exactly the same batches of pharmacological reagents and by the same experimenter, this might be mentioned explicitly as it further strengthens the point that the differences are genuine and not the result of differences in procedures or reagents.*

We initiated the rat experiments toward the end of the collection of data from human DRG neurons. However, the same stocks of test agents were used in both rats and humans. We have added these details to the revised manuscript.

*4) A missing piece of information is exactly how the authors selected which cells to record from. From the fact that they all had TTX-R current, the experimenter presumably chose smaller-diameter cells to record from, but what exactly was the selection criterion? Do the authors think the cells all correspond to cell bodies of C-fibers? The authors have discussed the challenges of high Na current density and series resistance that often plague studies of VGSCs. Because human DRG soma are much larger than rat, the demands on clamp quality would be systematically more challenging in recordings from human than rat neurons. This might cause a systematic difference in the apparent gating behavior. The distribution of cell capacitance did, however, have a region of overlap around 50 pF. Perhaps a subset of rat and human DRG neurons with comparable size could be analyzed to provide reassurance that the observed species differences were still detectable in a population of similar-sized neurons?*

We thank the reviewer for raising this issue. We did, in fact start this study with a focus on small to medium diameter human DRG neurons, based on our interest in pain and the rodent data indicating an association between afferent cell body diameter and function. However, when we noticed that the vast majority of even the medium diameter neurons also contained TTX-R currents, those generally associated with nociceptors, we also started to record from larger neurons. And while we generally avoided the largest of neurons because of the higher likelihood for larger currents and consequently greater difficulty with clamp control, in the end, we sampled a relatively broad distribution of cell body sizes, relative to the total distribution observed in cut sections. We have included data from cut sections in the revised manuscript. In doing so, we were able to expand on the implications of the cell body size distribution observed as well as the distribution of TTX-S currents with respect to our understanding of the relationship between cell body size and function in the Discussion of the revised manuscript.

While we have acknowledged the potential impact of clamp control problems on our characterization of current properties, we suggest errors associated with clamp control had a minimal impact on differences observed. This suggestion is based on fact that despite the more hyperpolarized voltage-dependence of activation of both TTX-R and TTX-S currents in human DRG neurons, there were only small differences in the slope of the G-V curves for the currents evoked from rat and human neurons.

Finally, with respect to the suggestion that it might be possible to record from rat and human neurons of comparable size, as shown in Figure 2, there was a small overlap in the size of the neurons studied from the two species. To address the reviewer’s concern, we have analyzed current properties of the largest rat neurons to compare to those in the smallest human neurons (in the 40-70 pF range). It should be acknowledged from the outset that this comparison is not ideal, as the largest rat neuron in which TTX-R currents were detected had a membrane capacitance of 54.3 pF. The observation that only TTX-S currents were detected in larger neurons is consistent with our previous experience and the suggestion that to the extent to which TTX-R currents are a reflection of neurons with nociceptive properties, this subpopulation of neurons tends to have a smaller cell body diameter. That said, even when these subpopulations of neurons were analyzed, differences in TTX-S and TTX-R currents in rat and human neurons were evident. For example, the V1/2 of current activation for TTX-R in the overlapping subpopulation of rat and human sensory neurons were -2.1 ± 0.4 mV and -10.8 ± 1.7 mV, respectively.

These details have been added to the revised manuscript.

5) The interpretation and discussion of VGSC deactivation was confusing (Figure 5, which shows plots of deactivation tau versus voltage). First, deactivation time constants are usually presented as a function of the deactivation pulse potential, not the preceding voltage to activate the channel as shown in Figure 5. However, from the text it seems that this is not a standard plot of tail time constant against voltage as one might assume at first but rather a measurement of the tail time constant at -60 mV following steps to different voltages. This should be made clear in the figure legend. Were there any measurements of deactivation time constants at other voltages? These would be helpful to someone who wanted to make a Hodgkin-Huxley type model from the data. Second, deactivation rates are usually measured over a range of test potentials. The more hyperpolarized ones are often regarded as more representative of "true" deactivation (O->C transition) but the fast kinetics are technically demanding. Responses at more depolarized potentials are more easily resolved, but may represent other state transitions (not just O -> C). The canonical view is that a true deactivation rate should be independent of the pulse potential used to open the channel. The atypical behavior in Figure 5 suggests a pulse to -60 mV was not measuring deactivation in isolation. Moreover, deactivation data are not presented for TTX-S currents. Perhaps the deactivation data should be omitted?

We regret that tail current data were not collected in a more traditional and appropriate manner. Unfortunately, we only realized this after the fact, despite the consistent observation that the TTX-R tail current in human DRG neurons looked consistently different from those evoked in the rat. Consequently, we attempted to describe the basis for this difference. Nevertheless, we agree with the reviewer that in the absence of a protocol enabling us to clearly describe deactivation, these data should be omitted and we have done so in the revised manuscript.

[Editors' note: further revisions were requested prior to acceptance, as described below.]

*The manuscript has been improved but there are some remaining issues that need to be addressed before acceptance, as outlined below:*

*"TTX-S" and "TTX-R" are used as shorthand for prepulse subtraction, which is validated for control solutions in Figure 1. This procedure may complicate the interpretation of some of the results, however, particularly the pharmacology leading to the conclusion that 1.7 is not the primary TTX-S channel. Please edit the text (especially the Introduction and Discussion) for the following:*

*1) To make sure that readers are clear that "TTX-S" and "TTX-R" actually refer to currents inactivated at -40 mV and available at -40 mV,*

We appreciate the reviewers concern. As this terminology is widely used throughout the literature to describe these two general current types in DRG neurons, that are not only readily distinguishable based on their sensitivity to TTX, but by their biophysical properties (both steady-state and kinetic) we have largely retained the use of the terminology in the Introduction. Although we did add a clause indicating that voltage- protocols were used in the majority of experiments described in this study. To further address the reviewer’s concern, however, we have included the following sentence in the Results section after the demonstration in Figure 1 that voltage and TTX can be used to isolate the same currents: “Based on these results, we refer to the slowly activating and slowly inactivation current resistant to steady-state inactivation as TTX-R current even though voltage steps rather than TTX was used to isolate this current in all subsequent experiments”. We go on clarify in the Discussion that we refer to this slow-activating and inactivating current in human DRG neurons as the TTX-R current, even though voltage was used for current isolation in all but the experiments described in Figure 1.

*2) To ensure that the prepulse subtraction procedure is kept in mind in interpreting the effects of the blockers, and*

We have included this caveat in the discussion of results obtained.

*3) To limit or constrain claims that NaV1.7 is not the primary TTX-S current, as appropriate.*

We have attempted to present the data for and against this proposal as objectively as possible. The possibility that NaV1.7 may not be the primary TTX-S current is suggested as an alternative possibility. We hope we have left it to the reader to decide, acknowledging that even our attempt at a balanced consideration of the data still favors this possibility.

*Please also include a statement about informed consent (as appropriate) for collection of human tissue.*

This information was included in the original version of our manuscript

*The full comments pertaining to the requested revisions are below and have been agreed upon by all participants in the review.*

*General assessment and major comments:*

*The most important change in the manuscript is new data with the Pfizer Nav1.7 inhibitor PF-05089771, which the authors find inhibits the TTX-S current quite potently, with an IC50 of about 10 nM. In addition, the authors have clarified some the details of the voltage protocols and cell selection.*

*I was very surprised that with the new data showing potent block of the TTX-S current by PF-05089771, the authors still end up concluding that Nav1.7 channels make little contribution to the TTX-S current in human DRG neurons. If it were my data, my interpretation of the overall set of the results the authors present would be the opposite, that the TTX-S current is likely to be primarily from Nav1.7 channels based on its sensitivity to PF-05089771, but that it is (interestingly) less sensitive to Protoxin-II than rat Nav1.7 channels. The observation that PF-05089771 also seems to inhibit TTX-R currents in human DRG neurons at higher concentrations is interesting and important. So, like most small molecules, the selectivity of PF-05089771 is imperfect. That is important to know, and constitutes another of the many interesting points in the manuscript about different pharmacology of the native currents in human DRG neurons compared to rat DRG neurons or heterologously-expressed human channels. However, it is still true that PF-05089771 inhibits the TTX-S current with significantly higher potency, and about what is expected from the effects on cloned human Nav1.7 channels. So my interpretation would be that the results with PF-05089771 imply that the TTX-S current is mostly from Nav1.7, but that the compound is less selective than previously realized and at higher concentrations also inhibits native human Nav1.8 channels – a very important point that adds to the significance of the results in the manuscript.*

We agree with the reviewer that one interpretation of our data is that NaV1.7 underlies the majority of TTX-S current in human DRG neurons. Our data with PF- 05089771 are indeed consistent with this interpretation. This is now clearly stated in the revised manuscript.

However, several lines of data support an alternative possibility, which is that NaV1.7 is not the primary subunit underlying TTX-S currents. First, while PF-05089771 was an order of magnitude more potent in the block of NaV1.7 than NaV1.2 or NaV1.6 expressed in HEK293 cells, with IC50 values of 0.011 μM, 0.11 μM and 0.16 μM respectively, our data suggest this compound is less selective against channels expressed in their native environment. That is, PF-05089771 had no activity at NaV1.8 channels expressed in HEK cells at concentrations as high as 10 μM, while TTX-R currents in human DRG neurons were blocked with an IC50 of ~60 nM. A comparable loss of selectivity at other channel subtypes would not enable PF-05089771 to differentiate between NaV1.7, NaV1.2 or NaV1.6. Importantly, semiquantitative PCR analysis of NaV subunit expression levels in whole mouse DRG indicates that NaV1.2 is mRNA is the most highly expressed TTX-S subunit (Laedermann et al., 2014). RNAseq analysis of DRG neurons suggests that the number of copies of NaV1.6 (31.1) is comparable to that of NaV1.7 (54), at least in the subpopulation of neurons not in the TRPV1 lineage (Goswami et al., 2014). Comparable results were obtained with a quantitative PCR analysis of single DRG neurons, where in those larger than 30 μm in diameter, NaV1.6 expression was comparable to that of NaV1.7 (Ho and O'Leary, 2011). Probably most importantly, there was only a 25% reduction in TTX-S current in DRG neurons from the NaV1.7 knock-out mouse (Nassar et al., 2004). Second, there was no evidence of a low threshold ramp current in human DRG neurons suggesting that if NaV1.7 was the dominant subunit, it would have very different biophysical properties than the current observed in rodent sensory neurons or in heterologous expression systems (Cummins et al., 1998). Third, there was no evidence of Protoxin II block of the TTX-S current in human DRG neurons suggesting that if NaV1.7 was the dominant subunit, it would also have very different toxin sensitivity than that of NaV1.7 expressed in heterologous expression systems (Schmalhofer et al., 2008) or in rodent sensory neurons (Laedermann et al., 2013). And fourth, inflammatory mediator-induced modulation of TTX-S currents was only observed in a subpopulation of neurons in which TTX-S currents were observed. While this could be explained by differences in second messenger signaling between neurons, it may also reflect a differential expression of TTX-S channel subtypes.

*To me, the flow of the revised manuscript now seems strange in that final set of experimental results in Results showing potent block by PF-05089771 gives the reader the impression that the TTX-S current is likely mainly Nav1.7, but then in the Discussion the first mention of this issue suggests the opposite without explaining why. The first part of the Discussion summarizing the pharmacology is very reasonable and straightforward (the paragraph starting, "With respect to drug development, the results of the present study are consistent with those from previous studies indicating that both expression systems and species differences may have a significant influence on the pharmacological properties of the protein in question." This paragraph nicely summarizes why it is not shocking that the pharmacology of channels might differ between rat and human cells and between native currents and those from heterologous expression.*

*The first suggestion in the Discussion that the authors conclude that Nav1.7 does not make the major contribution to TTX-S current comes obliquely, in a sentence in a paragraph about the up-regulation of currents by the inflammatory soup "As noted above, data from rodent DRG neurons suggest that NaV1.7 accounts for ~70% of the TTX-S current in small diameter neurons and NaV1.6 accounts for the remainder (Laedermann et al., 2013). However, the suggestion that the contribution of NaV1.7 varies across subpopulations of human sensory neurons is also consistent with the suggestion that NaV1.7 is not the dominant, let alone only VGSC subtype in human sensory neurons." In the original version, this sentence referred to a stated conclusion in Results based on the protoxin-II results that Nav1.7 was not the dominant current. But in the revised version, there has been no explanation yet of why the authors would conclude this, and it seems to come out of nowhere.*

We have reworked the Discussion to address the reviewers concerns. The suggestion that NaV1.7 might not be the dominant subunit in human DRG neurons is now only raised in the context of a Discussion of the possibility that NaV1.7 may not be the dominant subunit underlying the TTX-S currents in human DRG neurons. And this possibility is only raised after a Discussion of the possibility that NaV1.7 does underlie the TTX-S current.

*The authors' argument about Nav1.7 comes later, in the paragraph starting "The possibility that NaV1.7 may not be the dominant subunit underlying TTX-S currents, let alone significantly contribute to these currents in human DRG neurons, as suggested by our inflammatory mediator results, was further supported by our results with Pro-Tx II and PF-05089771." This is where the authors argue that the block by PF-05089771 is not good evidence for Nav1.7 mediating the TTX-S because they find that PF-05089771 at higher concentrations also inhibits the TTX-R current. To me, the argument in this paragraph was not convincing, as it essentially argues that the lack of effect of protoxin-II is more convincing than the potent block by PF-05089771. (The lack of ramp currents from TTX-S currents is another argument, but as the authors point out when presenting this data, this could be easily explained by different gating kinetics between human and rat TTX-S currents.).*

We appreciate the reviewers concerns and agree that this argument could and should have been made better on the one hand, and that the possibility that NaV1.7 could still be the dominant subunit made more clearly on the other. With respect to the reviewers primary concern about the selectivity of PF-05089771, we acknowledge that it is possible that the compound retained its selectivity for NaV1.7 over NaV1.2 and NaV1.6 (as well as the rest of the TTX-S subunits present in sensory neurons), while dramatically losing selectivity for NaV1.8 in human DRG neurons. However, a greater than three orders of magnitude increase in potency would at least raise the possibility that the compound loses all selectivity against human channels in their native environment.

*There is a reason to be cautious about interpreting the results with PF-05089771 and Protoxin-II, because both interact with their target channels in a state-dependent manner. The authors did not study the effects of the blockers on isolated currents from TTX-R or TTX-S channels, but rather used a prepulse procedure to distinguish them by inactivation. They show that this protocol works in the absence of drugs, but obviously it may not in the presence of drugs that can modify the voltage-dependence of inactivation. In my view, that should introduce a lot of caution in interpreting these results.*

We completely agree with the reviewer that it is important to consider the potential implications of state-dependent block. However, we disagree with the implication that the voltage protocol used to isolate TTX-R from TTX-S could account for the observation that TTX-R currents were blocked by PF-05089771. This compound blocks inactivated channels more potently that channels in a resting or closed state. The protocol used to evoke TTX-R currents should have minimized any inactivated-state block of these channels, and consequently any detectable block of these channels.

Furthermore, the TTX-R currents remaining in the presence of PF-05089771 had biophysical properties comparable to the TTX-R currents recorded in the absence of the compound, with a comparable voltage-dependence of inactivation, as well as kinetics of activation and inactivation. Conversely, any contamination of TTX-R currents observed in the presence of PF-05089771 by TTX-S currents whose gating properties were somehow modified by the presence of the compound would suggest that the potency of the compound was lower in human DRG neurons than in HEK cells.

*In fact, in reading the paper quickly, it would be very easy for a reader to miss the important point that when the authors refer to "TTX-S" and "TTX-R" currents, this is really shorthand for "currents available from -80 but inactivated by a prepulse to -40 mv" and "currents remaining after a prepulse to -40 mV". Although the authors clearly state this in the course of Results, I think it is an important point to reiterate in the Discussion, because it is an important qualification for interpreting the pharmacological results (not only for protoxin and PF-05089771, but also lidocaine and A-803467, which are also state-dependent blockers).*

We completely agree with the reviewer and have reiterated this point in the Introduction, Results, and Discussion as suggested.

*Because of these issues, I did not find the authors' suggestion that Nav1.7 accounts for little if any of the TTX-S current convincing and it would not be my interpretation. Of course, the authors should be free to interpret their own data, but they should probably consider at least qualifying their conclusion.*

As noted above, we have tried to lay out the evidence both for and against the contribution of NaV1.7 to the TTX-S currents in human DRG neurons more clearly.

*Whatever the interpretation given, the experimental results are very interesting and leave no doubt that the detailed pharmacology of the channels in human cells is quantitatively different in interesting ways compared to the currents in rat neurons, probably most importantly in the relative lack of use-dependence with lidocaine and the apparent significant sensitivity of TTX-R channels to PF-05089771.*

We appreciate the supportive comments.

I did not see a statement about informed consent for the collection of the human tissue, which seems important. Presumably this is the authors' 2015 paper that is cited for the Methods, but it seems that it should be here also.

We agree with the reviewer that this is an important point. This information was in the first sentence of Materials and methods section of this manuscript.